# Cell-type-specific DNA methylation dynamics in the prenatal and postnatal human cortex

## Graphical abstract

## Authors

Alice Franklin, Jonathan P. Davies, Nicholas E. Clifton, ..., Emma Dempster, Eilis Hannon, Jonathan Mill

## Correspondence

a.franklin@exeter.ac.uk (A.F.), j.mill@exeter.ac.uk (J.M.)

## In brief

Franklin et al. profile DNA methylation during development and aging of the human cortex, identifying pronounced cell-type-specific changes during prenatal life. The authors confirm the importance of epigenetic mechanisms in establishing cell-type-specific signatures during human cortex development and link dynamic developmental methylation patterns to genes implicated in autism and schizophrenia.

## Highlights

- Dynamic DNA methylation changes shape the developing human cortex

- Prenatal changes in DNA methylation are distinct from postnatal aging patterns

- Evidence for neuron-specific DNA methylation changes in the developing cortex

- Autism and schizophrenia genes are enriched for developmental DNA methylation changes

Franklin et al., 2025, Cell Genomics 5, 101010
December 10, 2025 © 2025 The Author(s). Published by Elsevier Inc.

# Cell Genomics

CellPress

## Resource

# Cell-type-specific DNA methylation dynamics in the prenatal and postnatal human cortex

Alice Franklin,[1,*] Jonathan P. Davies,[1] Nicholas E. Clifton,[1] Georgina E.T. Blake,[1] Rosemary Bamford,[1] Emma M. Walker,[1] Barry Chioza,[1] Martyn Frith,[1] APEX Consortium, Youth-GEMs Consortium, Joe Burrage,[1] Nick Owens,[1] Shyam Prabhakar,[2] Emma Dempster,[1] Eilis Hannon,[1] and Jonathan Mill[1,3,*]

[1]Department of Clinical & Biomedical Sciences, University of Exeter Medical School, Exeter, UK
[2]Laboratory of Systems Biology and Data Analytics, Genome Institute of Singapore, Singapore, Singapore
[3]Lead contact
*Correspondence: a.franklin@exeter.ac.uk (A.F.), j.mill@exeter.ac.uk (J.M.)

## SUMMARY

The human cortex undergoes extensive epigenetic remodeling during development, although the precise temporal and cell-type-specific dynamics of DNA methylation remain incompletely understood. In this study, we profiled genome-wide DNA methylation across human cortex tissue from donors aged 6 post-conception weeks to 108 years of age. We observed widespread, developmentally regulated changes in DNA methylation, with pronounced shifts occurring during early- and mid-gestation that were distinct from age-associated modifications in the postnatal cortex. Using fluorescence-activated nuclei sorting, we optimized a protocol for the isolation of SATB2-positive neuronal nuclei, enabling the identification of cell-type-specific DNA methylation trajectories in the developing cortex. Developmentally dynamic DNA methylation sites were significantly enriched near genes implicated in autism and schizophrenia, supporting a role for epigenetic dysregulation in neurodevelopmental conditions. Our findings underscore the prenatal period as a critical window of epigenomic plasticity in the brain with important implications for understanding the genetic basis of neurodevelopmental phenotypes.

## INTRODUCTION

Development of the human cortex is a complex and highly orchestrated process underpinned by the temporally coordinated regulation of gene expression.[1] Starting with the rapid proliferation of neural progenitor cells in the ventricular zone, these transcriptional programs drive key developmental processes, including neurogenesis and synaptogenesis, and act to regulate neuronal growth, migration, and connectivity in the developing cortex. Importantly, dysregulation to these pathways is implicated in neurodevelopmental disorders, including autism and schizophrenia.[2,3] The cortex continues to develop throughout fetal life and into the postnatal period, when synaptic pruning and myelination further refine neural circuits during childhood and adolescence.[4]

Epigenetic modifications are essential for the dynamic regulation of gene function during cellular differentiation in the developing central nervous system. The most studied epigenetic mechanism is DNA methylation, which involves the addition of a methyl group to the fifth carbon of cytosine. DNA methylation is primarily thought to inhibit local gene expression by disrupting transcription factor binding and attracting methyl-binding proteins that promote chromatin compaction and gene silencing.[5] However, its effects on transcription can vary depending on the genomic and cellular context. For instance, DNA methylation within the gene body is often linked to increased gene expres-

sion[6] and has been associated with other genomic processes, such as alternative splicing and promoter usage.[7] Epigenetic processes are central to transcriptional plasticity in the developing central nervous system.[8] DNA methylation plays a crucial role in neurodevelopment, as shown by the dynamic expression of the *de novo* DNA methyltransferases DNMT3A and DNMT3B in the developing brain.[9] Mutations in the methyl-CpG-binding protein 2 (MECP2) gene, which regulates neuronal gene expression by interacting with methylated DNA, cause severe neurodevelopmental deficits.[10] Moreover, DNA methylation is known to be involved in key neurobiological and cognitive functions throughout life, including neuronal plasticity,[8] memory formation and retention,[11] and circadian rhythms.[12] Current analyses of DNA methylation in the developing cortex have profiled tissue from a small number of donors spanning a narrow range of developmental ages,[13–15] and little is known about the extent to which these changes continue during later stages of development and during postnatal life. Importantly, given the role of DNA methylation in establishing cellular identity, existing studies have not systematically explored developmental patterns of DNA methylation in specific cell populations.

In this study, we quantified DNA methylation across the genome in fetal cortex, reporting dramatic changes in DNA methylation across development that are specific to the early- and mid-gestational period and distinct from age-associated changes observed in late gestation and postnatally. We

developed a fluorescence-activated nuclei sorting (FANS) protocol to isolate SATB2-positive neuronal nuclei and identify cell-type-specific trajectories of DNA methylation. Finally, we show that neurodevelopmentally dynamic DNA methylation sites are enriched in the vicinity of genes implicated in autism and schizophrenia, supporting a role for neurodevelopmental processes in these conditions. This is, to our knowledge, the most extensive study of DNA methylation across development of the human cortex and confirms the prenatal period as a time of considerable epigenomic plasticity.

## RESULTS

### Dramatic changes in DNA methylation during development of the human cortex

Our first analyses focused on characterizing changes in DNA methylation during early- and mid-gestation fetal cortex development. Using cortex tissue from 91 fetal donors (age range = 6–23 post-conception weeks [pcw], male = 45 [age range = 6–20 pcw], female = 46 [age range = 8–23 pcw]; Table S1), DNA methylation was quantified across the genome using the Illumina EPIC microarray (see STAR Methods). After stringent pre-processing, our final dataset included DNA methylation data for 807,806 sites (789,981 autosomal sites and 17,626 on the X chromosome). Using an epigenetic clock calibrated specifically for fetal brain,[16] we found the expected strong correlation between predicted and actual developmental age (correlation [corr] = 0.942, Figure S1). We fitted a linear model of DNA methylation level as a function of developmental age (in pcw) controlling for sex and experimental batch (see STAR Methods) and identified widespread changes in DNA methylation associated with developmental age, finding 50,913 (6.30% of total) differentially methylated positions associated with cortex development (dDMPs) (50,329 [98.9%] autosomal, 572 [1.12%] X chromosome) at an empirically derived experiment-wide significance threshold ($p < 9 \times 10^{-8}$)[17] (Table S2). Effect sizes for dDMPs that overlapped with sites reported in two previous studies of the developing fetal brain performed using older technology were highly correlated across studies (Numata et al.[15]: number of overlapping sites = 66, corr = 0.94. Spiers et al.[13]: number of overlapping sites = 20,503, corr = 0.824; Figure S2). Consistent with our previous results, we found a small but highly significant enrichment of dDMPs becoming hypomethylated with cortex development ($n = 28,780$ [56.5%], $p = 1.61 \times 10^{-191}$) with the mean effect size being significantly greater for hypomethylated dDMPs than hypermethylated dDMPs (change in DNA methylation [%] per week: hypermethylated dDMPs = 1.03, hypomethylated dDMPs = −1.53, t test $p < 1 \times 10^{-320}$, Figure S3). Reflecting this, global levels of DNA methylation, calculated by taking the mean across all autosomal DNA methylation sites included in the final dataset, became slightly, but significantly, lower across fetal development (change in DNA methylation [%] per week = −0.0194, $p = 3.33 \times 10^{-10}$) (Figure S4).

The top-ranked dDMP (cg08125539), located within the gene encoding insulin-like growth factor 2 mRNA-binding protein 1 (*IGF2BP1*), was characterized by a rapid increase in DNA methylation in the developing cortex (increase in DNA methylation [%] per pcw = 4.46, SE = 0.183, $p = 1.16 \times 10^{-40}$; Figure 1A). *IGF2BP1*

plays a critical role in the development of the human cortex, being highly expressed during early embryogenesis but silenced later in gestation and postnatally.[18] The top-ranked dDMP becoming hypomethylated across cortex development was cg11884704, located within the gene encoding solute carrier family 25 member 25 (*SLC25A25*) (change in DNA methylation [%] per week = −2.71, SE = 0.114, $p = 8.63 \times 10^{-40}$; Figure 1B). Many of the dDMPs associated with cortex development were spatially co-located, clustering into developmentally differentially methylated regions (dDMRs) spanning multiple DNA methylation sites. Using *dmrff*,[19] we identified 1,356 dDMRs (corrected $p < 0.05$, number of dDMPs $\geq 3$, mean size = 369 bp) associated with cortical development, annotated to 2,292 genes (Table S3). Many of the top-ranked dDMRs are proximal to genes with established roles in development and function of the cortex. For example, the most significant dDMR, characterized by a dramatic increase in DNA methylation across cortex development (chromosome 11: 31,846,414–31,849,262, mean change in DNA methylation [%] per week across region = 6.50, $p < 1 \times 10^{-320}$) spans 20 sites overlapping a CpG island in *PAX6* (Figure 1C), a gene encoding a potent brain-expressed transcription factor with a critical role in neurogenesis and cortical development.[20]

### DNA methylation at a large proportion of sites changes nonlinearly across cortex development

The regression model used to identify dDMPs was unable to distinguish between linear and nonlinear changes in DNA methylation in the developing cortex. Nonlinear changes could provide important insights into epigenetic switch points during key stages of brain development,[22] and therefore we sought to apply non-parametric modeling to identify nonlinear trajectories of DNA methylation. Using Gaussian process regression and model selection, we classified DNA methylation sites as constant, linear, or nonlinear throughout early- and mid-fetal cortex development. To reduce the chance of falsely identifying nonlinear patterns, we excluded non-variable probes, removed outlier samples, and focused on sites where DNA methylation changed within a biologically relevant time frame (see STAR Methods). We identified 73,035 sites (72,310 [99.0%] autosomal) that demonstrated high-confidence nonlinear changes in DNA methylation during the fetal period (Table S4). This includes 23,252 (45.7%) of the dDMPs detected by our linear model (Figure S5A), highlighting that the rate of change in DNA methylation at these sites is not consistent across development (Figures S5B and S5C). We applied weighted gene correlation network analysis (WGCNA)[23] to group autosomal sites characterized by shared nonlinear DNA methylation changes, finding six distinct modules (Figure 2; Table S4). Of note, Gene Ontology analysis of genes annotated to sites in each module revealed relevant biological functions. For example, the turquoise module was enriched for relatively broad developmental processes (e.g., "system development," $p = 7.81 \times 10^{-17}$), cell signaling ("cell-cell signaling," $p = 4.74 \times 10^{-14}$), and the extracellular matrix (e.g., "actin cytoskeleton organization," $p = 3.55 \times 10^{-11}$), while the blue module was enriched for pathways more specifically related to synaptic structure and signaling (e.g., "synapse," $p = 4.24 \times 10^{-13}$, and "neuron projection," $p = 4.09 \times 10^{-11}$) (Table S5).

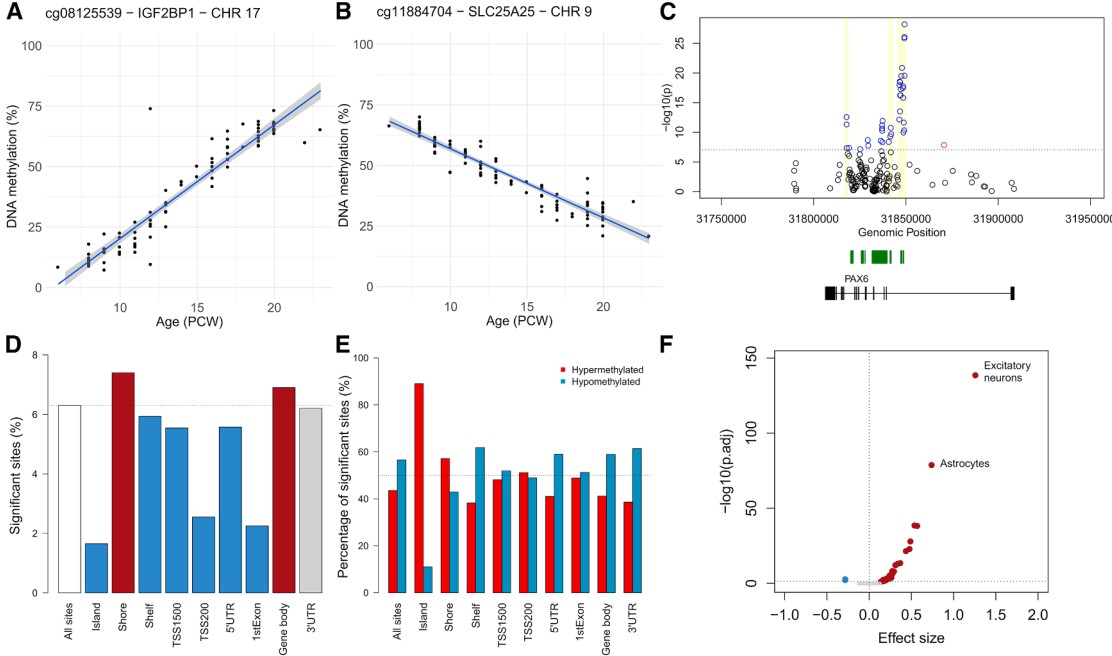

**Figure 1. Development-associated changes in DNA methylation in the human fetal cortex**

(A and B) The top-ranked dDMPs demonstrating (A) hypermethylation and (B) hypomethylation over fetal cortex development. cg08125539, annotated to *IGF2BP1* on chromosome 17, is characterized by a significant increase in DNA methylation across prenatal cortex development (% increase in DNA methylation = 4.5% per week, $p = 1.16 \times 10^{-40}$). cg11884704, annotated to *SLC25A25* on chromosome 9, is characterized by a significant decrease in DNA methylation across prenatal cortex development (% decrease in DNA methylation = 2.7% per week, $p = 8.63 \times 10^{-40}$). Of note, the dramatic changes in DNA methylation at these sites are specific to the prenatal period (see also Figure S11). Shaded region indicates 95% confidence interval.

(C) dDMPs cluster into differentially methylated regions (DMRs) associated with cortex development (see also Table S3). The *PAX6* gene on chromosome 11 contains three independent DMRs (yellow shaded regions), including the top-ranked DMR (mean change in DNA methylation % per week across region = 6.50, $p < 1 \times 10^{-320}$) overlapping an intragenic CpG island (green). Blue, dDMP with a positive effect size; red, dDMP with a negative effect size; black, non-significant site. Dotted line indicates genome-wide significance.

(D) Relative enrichment (red) and depletion (blue) of dDMPs across different CGI and genic features. Compared to the frequency of dDMPs among all sites profiled in this study (white bar), there was a significant enrichment of dDMPs in CGI shores and gene bodies but a significant depletion in other regions, most notably CGIs (see also Table S7).

(E) Proportion of hypermethylated (blue) and hypomethylated (red) dDMPs across CGI and genic features. Despite the overall enrichment of hypomethylated dDMPs across all sites tested, there are substantial differences across genomic features with CGIs and CGI shores characterized by an enrichment of hypermethylated dDMPs (see also Table S8).

(F) Enrichment of cortex dDMPs in cell-type-specific regions of open chromatin identified by scATAC-seq.[21] Shown is a volcano plot of the relative effect size versus *p* value from a logistic regression testing for an enrichment of dDMPs within cell-type-specific scATAC-seq peaks for 54 human fetal cell types. Peaks for 31 fetal cell types were significantly enriched for dDMPs, highlighting a general overlap with transcriptionally active regions of the genome during development (red, significant enrichment; blue, significant depletion; gray, non-significant). The greatest enrichment was observed within ATAC-seq peaks specific to fetal excitatory neurons (effect size = 1.27, odds ratio = 3.56, corrected $p = 1.17 \times 10^{-139}$) and astrocytes (effect size = 0.75, odds ratio = 2.12, corrected $p = 6.74 \times 10^{-80}$).

## dDMPs are enriched in specific genomic features and developmentally active regions of open chromatin

Although the dDMPs identified using our linear regression model were distributed relatively equally across autosomal chromosomes, certain chromosomes were characterized by a relative enrichment or depletion (Figure S6; Table S6), most notably chromosome 19 (percentage of dDMPs = 3.49%, log odds ratio = −1.81, corrected $p = 2.16 \times 10^{-109}$); of note, this chromosome has the highest gene density of any human chromosome.[24] The distribution of dDMPs across genic features was more dramatically skewed (Figure 1D; Table S7); for example, there was a significant depletion of dDMPs in promoter regulatory regions including CpG islands (CGIs) (percentage of dDMPs =

1.65%, log odds ratio = −3.81, corrected $p < 1 \times 10^{-320}$). In contrast, dDMPs were enriched in CGI shores (percentage of dDMPs = 7.40%, log odds ratio = 1.17, corrected $p = 1.74 \times 10^{-79}$). dDMPs located in different genic features were also enriched for sites becoming either hypo- or hypermethylated during cortex development (Figure 1E; Table S8). Despite the genome-wide enrichment of hypomethylated dDMPs, specific features were characterized by a dramatic enrichment of hypermethylated dDMPs, including CGIs (hypermethylated dDMPs = 89.0%, hypomethylated dDMPs = 11.0%, corrected $p < 1 \times 10^{-320}$) and CGI shores (hypermethylated dDMPs = 57.1%, hypomethylated dDMPs = 42.9%, corrected $p = 1.50 \times 10^{-48}$). Sites characterized by nonlinear

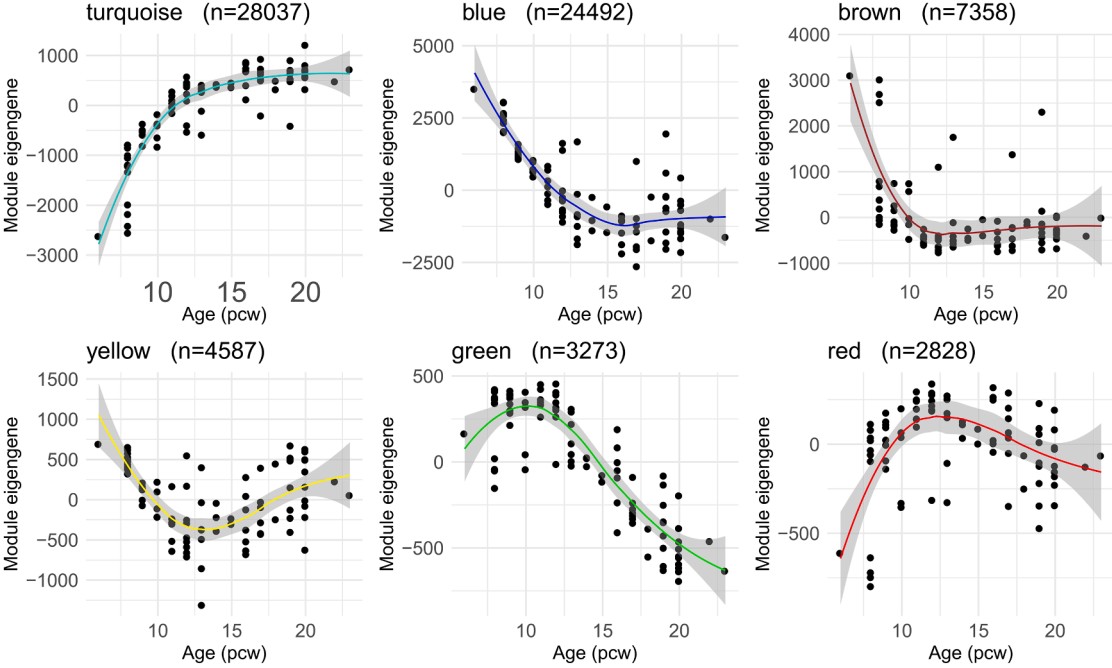

**Figure 2. Nonlinear trajectories of DNA methylation in the developing cortex**
Nonlinear DNA methylation sites identified through Gaussian process regression modeling were clustered into distinct modules using WGCNA (STAR Methods). For each module, the eigengene (first PC) is plotted against fetal age (pcw). The number of nonlinear sites assigned to each module is indicated in the subtitles, with hub DNA methylation sites for each module shown in Figure S29.

changes in DNA methylation across cortex development showed a very similar distribution across chromosomes and genic features to dDMPs identified using our linear model, with, for example, a highly significant depletion of developmentally dynamic sites in CGIs (background percentage nonlinear = 13.7%, percentage nonlinear sites annotated to CGI = 6.94%, log odds ratio = −1.98, corrected $p < 1 \times 10^{-320}$). These patterns resemble those seen in the development of other tissues[25] and reflect the observation that CGIs typically exhibit low DNA methylation levels early in development, with tissue-specific DNA methylation patterns becoming established during the prenatal period.[26]

Using publicly available single-cell ATAC-seq (scATAC-seq) data generated from multiple human fetal tissues,[21] we explored the extent to which autosomal dDMPs (n = 50,329) were enriched in regions of open chromatin associated with the development of different cell types. dDMPs were significantly enriched in the top 10,000 ATAC-seq peaks identified in 31 out of 54 fetal cell types tested, suggesting a broad overlap with transcriptionally active regions of the genome during development (Table S9). Strikingly, however, the greatest enrichment was observed within ATAC-seq peaks specific to fetal excitatory neurons (effect size = 1.27, odds ratio = 3.56, corrected $p = 1.17 \times 10^{-139}$) and astrocytes (effect size = 0.75, odds ratio = 2.12, corrected $p = 6.74 \times 10^{-80}$) (Figures 1F and S7A). Collectively, we found that nonlinear sites again exhibited a similar pattern of enrichment within cell-type-specific ATAC-seq peaks, although individual nonlinear WGCNA modules showed enrichment for different cell types (Figure S7B). For example, the

blue module was strongly enriched for excitatory neurons (effect size = 2.58, odds ratio = 13.2, corrected $p = 5.40 \times 10^{-319}$) while the brown module showed the strongest enrichment for astrocytes (effect size = 1.60, odds ratio = 4.95, corrected $p = 2.48 \times 10^{-161}$).

## Neurodevelopmental changes in DNA methylation are largely distinct from those occurring with age in the postnatal cortex

Given the major shifts in DNA methylation observed during cortex development, we were interested in characterizing the extent to which these changes were maintained across the life course. We quantified DNA methylation in cortex tissue from postnatal donors (n = 673, age range = 0–104 years; Table S1) in addition to a small number of late-fetal cortex samples (n = 4, age range = 26–33 pcw; Table S1). As expected, chronological age was strongly correlated with age estimates derived from DNA methylation data using both a pan-tissue epigenetic clock[27] (corr = 0.875; Figure S8A) and a clock trained specifically on postnatal human cortex[28] (corr = 0.941; Figure S8B). Overall, DNA methylation at dDMPs was dramatically more variable across early- and mid-fetal cortex samples than adult cortex samples (Figure 3A) (mean variance in DNA methylation [%] across the top 10,000 dDMPs profiled in both fetal and adult cortex = 120 and 41.7, respectively, t test $p < 1 \times 10^{-320}$). We next tested the extent to which development-associated DMPs were also characterized by age-associated changes in DNA methylation in the postnatal cortex, finding that, for the 41,518 dDMPs also tested in our postnatal cortex samples (81.5% of all dDMPs), there was

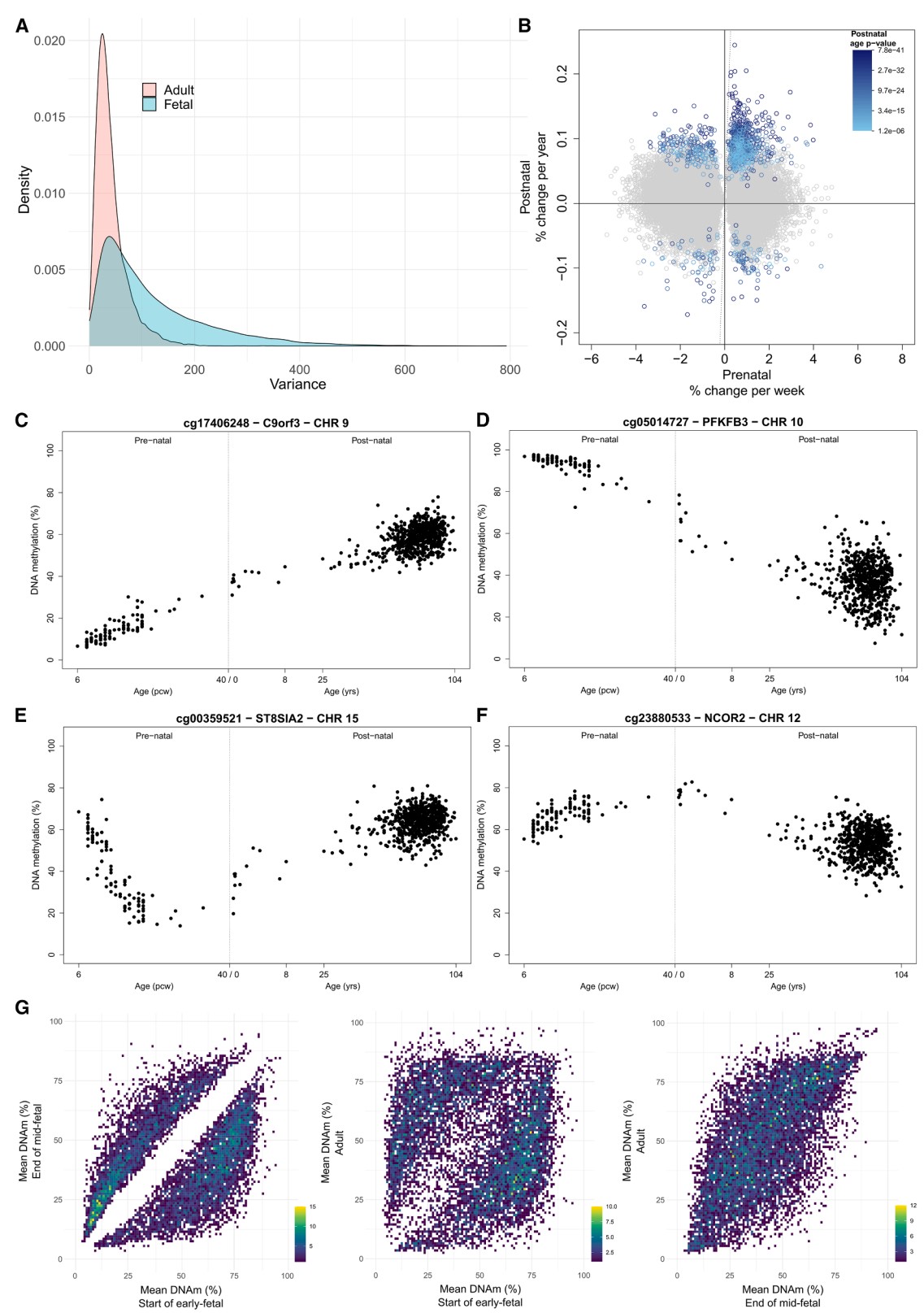

*(legend on next page)*

minimal correlation in age effect size between prenatal and postnatal cortex (corr = −0.00925) (Figure 3B). DNA methylation at only a very small proportion of these sites ($n = 1,003$ [2.42%]) was significantly ($p < 1.20 \times 10^{-6}$) associated with age in postnatal samples, with 275 (27.4%) of these sites being characterized by age-associated changes in the *opposite* direction to those observed in the developing cortex (Figures 3C–3F).

Of note, dDMPs that become hypermethylated across cortex development showed a higher proportion of consistent postnatal age effects ($n = 664$ [89.4% of the 743 hypermethylated dDMPs that are also significant postnatally]) compared to sites becoming hypomethylated across cortex development ($n = 124$ [24.6% of the 260 hypomethylated dDMPs that are also significant postnatally]). Furthermore, while there is an overall depletion of dDMPs in CGIs, dDMPs located in CGIs were characterized by the most consistent age-associated changes in DNA methylation between prenatal and postnatal cortex (Figure S9). Among dDMPs also associated with age in postnatal cortex ($n = 1,003$), there was a significant enrichment for sites located in CGIs (relative enrichment = 3.67, corrected $p = 3.97 \times 10^{-99}$) (Figure S10). Together, these findings suggest that CpG islands may be preferentially involved in general aging-related methylation changes, whereas dDMPs in other genomic contexts (e.g., shores and shelves) are more likely to reflect development-specific epigenetic remodeling.

For the majority of cortex dDMPs, mean levels of DNA methylation in adult cortex ($n = 661$, age range = 25–104 years [mean age = 81.8 years]) were highly correlated to the mean DNA methylation of the oldest fetal samples ($n = 12$ donors, aged 20–23 pcw) and dramatically different from that of the youngest fetal samples (21 donors, aged 6–9 pcw) (Figure 3G). Indeed, the top hyper- and hypomethylated dDMPs (cg08125539 [annotated to *IGF2BP1*] and cg11884704 [annotated to *SLC25A25*], respectively) show little variation in DNA methylation level after birth (Figure S11), exemplifying the relative stability of DNA methylation postnatally following the dramatic developmental shifts observed in the prenatal cortex.

### Developing a method to profile DNA methylation in excitatory neurons in the developing cortex

Different neural cell types are characterized by distinct transcriptional trajectories across development of the human cortex.[29] As the analysis of bulk cortex precludes the ability to identify changes in DNA methylation occurring in specific cell types and might be confounded by the shifts in cell-type proportions

occurring during this period,[30] we next sought to isolate nuclei from developing neurons prior to methylomic profiling. The nuclear marker NeuN (encoded by the *RBFOX3* gene) is widely used to label neuronal nuclei prior to FANS and genomic profiling in postnatal human cortex,[31] and we attempted to use this approach to profile DNA methylation in cortical NeuN+ and NeuN− nuclei populations obtained from donors spanning the life course (see STAR Methods). Despite NeuN robustly identifying discrete neuronal nuclei populations in late-fetal and postnatal cortex samples, we found that it was not a reliable nuclear neuronal lineage marker in early- and mid-fetal donors. Although *RBFOX3* is expressed in postmitotic neurons and plays a role in neuron differentiation,[32] interrogation of RNA-seq data generated by the BrainSeq Consortium[33] shows its expression is dramatically lower in fetal cortex compared to *SATB2* (special AT-rich sequence-binding protein 2), a potent transcription factor that drives early neurogenesis in the developing cortex and is postnatally expressed in excitatory neurons[34] ($t = -14.6$, $p < 1 \times 10^{-320}$; Figure S12). We therefore optimized a method to immunolabel SATB2+ nuclei prior to FANS purification, using snRNA-seq to confirm the co-expression of *RBFOX3* and *SATB2* in postnatal excitatory neurons (Figure S13) and the higher expression of *SATB2* compared to *RBFOX3* in SATB2+ nuclei isolated from the fetal cortex (Figure S14). Our protocol for the isolation of SATB2+ nuclei from fetal cortex is available as a resource on Protocols.io (https://doi.org/10.17504/protocols.io.n92ldz9d8v5b/v1).

### Differences in neuronal DNA methylation patterns between fetal and adult cortex

We used FANS to isolate SATB2+ (neuron-enriched) and SATB2− (neuron-depleted) nuclei populations from a subset ($n = 37$) of prenatal cortex tissue samples (Table S1), profiling DNA methylation in each purified fraction as previously described (see STAR Methods). Of note, although we confirmed that the purity of the SATB2+ isolated nuclei population is high (Figure S14), the neuron-depleted population is more heterogeneous and comprises a mix of different cell types, including some inhibitory neurons that do not express SATB2. These data were combined with existing DNA methylation data generated on neuron-enriched (NeuN+), oligodendrocyte-enriched (SOX10+), and microglia-enriched (IRF8+ and NeuN−/SOX10−) nuclei from 212 postnatal cortex samples by our group, resulting in a cell-type-specific DNA methylation dataset from 259 donors aged 8 pcw to 108 years (Table S1). Principal

**Figure 3. The majority of dDMPs are not characterized by age-associated changes in DNA methylation in the postnatal cortex**
(A) DNA methylation at dDMPs was dramatically more variable in fetal cortex samples than adult cortex samples. Shown is the distribution of variance in DNA methylation across the 10,000 most significant dDMPs in fetal samples (pink) (mean variance = 120, mean SD = 11% DNA methylation) compared to the variance in DNA methylation at the same sites in postnatal samples (blue) (mean variance = 41.7, mean SD = 6.46% DNA methylation).
(B–F) (B) Comparison of age effect sizes between prenatal and postnatal cortex for the 41,518 dDMPs also measured in the postnatal samples. Each dot represents a cortex dDMP with significance for age in the postnatal cortex indicated by color (gray, not significantly associated with age in postnatal cortex). Examples of dDMPs showing (C) a consistent age-associated increase in DNA methylation in both prenatal and postnatal cortex, (D) a consistent age-associated decrease in DNA methylation in both fetal and postnatal cortex, (E) a developmentally associated decrease in DNA methylation in the prenatal cortex followed by an age-associated increase in DNA methylation in the postnatal cortex, and (F) a developmentally associated increase in DNA methylation in the prenatal cortex followed by an age-associated decrease in DNA methylation in the postnatal cortex.
(G) Comparisons of mean DNA methylation at the 10,000 top-ranked dDMPs between the earliest fetal samples ($n = 21$, age range = 6–9 pcw), the eldest mid-fetal samples ($n = 12$, age range = 20–23 pcw), and adult samples ($n = 661$, age range = 25–104 years). Mean DNA methylation of the mid-fetal samples is most strongly correlated to postnatal samples (corr = 0.65) (right) than to early-fetal samples (corr = 0.36) (left). Color bar indicates number of DNA methylation sites per bin.

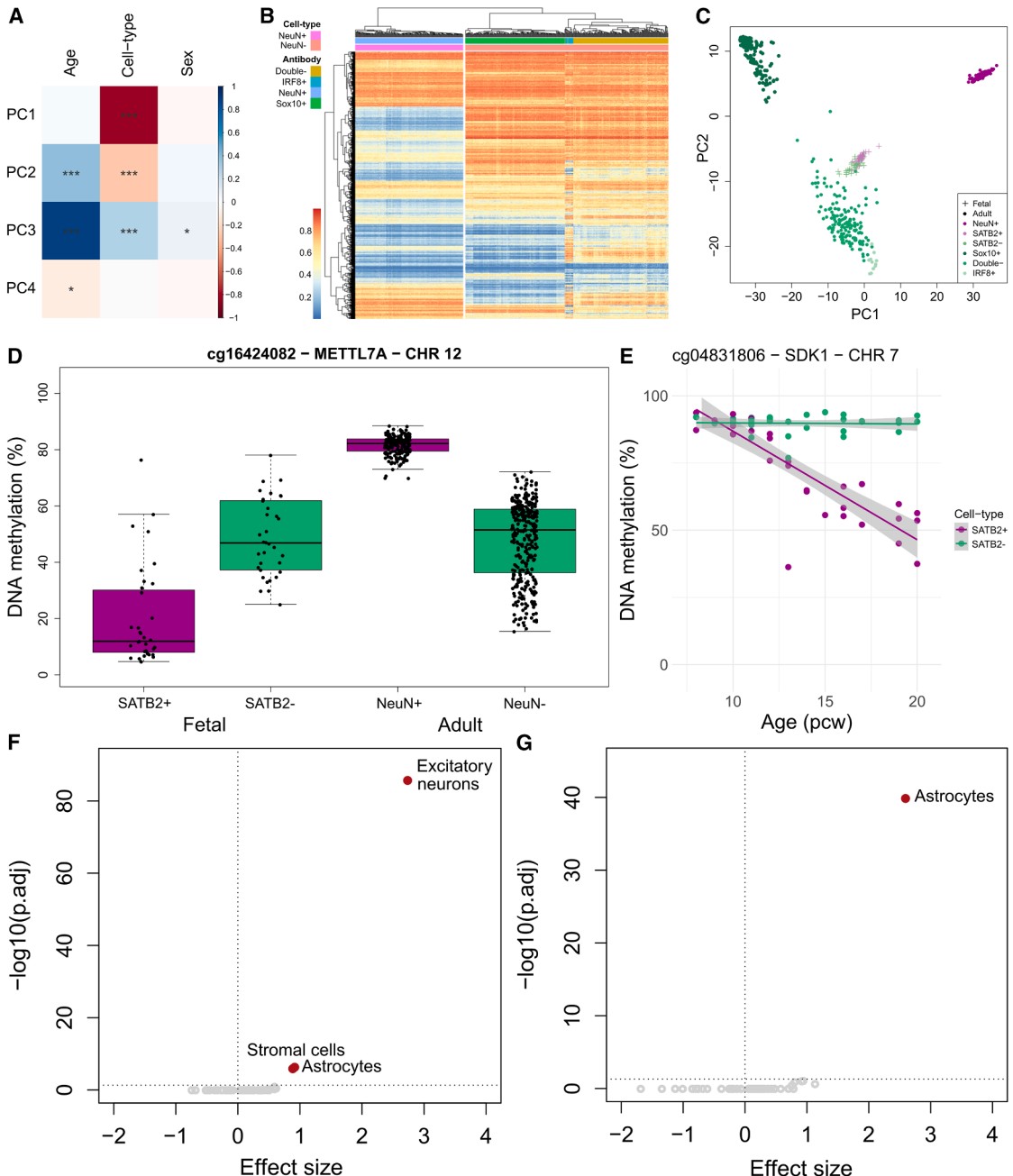

**Figure 4. Cell-type-specific changes in DNA methylation across cortex development and postnatal life**

(A) Principal component analysis (PCA) on the top 10,000 most variable autosomal DNA methylation sites across FANS-isolated nuclei populations from prenatal and postnatal cortex samples ($n$ = 259 donors). The top four principal components (PCs) correlated against age, cell type, and sex (*$p$ < 0.05, ***$p$ < 0.001).

(B) Heatmap of DNA methylation in purified nuclei populations isolated from adult cortex at the 6,531 DMPs between SATB2+ and SATB2− nuclei identified in fetal cortex.

(C) PCA on the top 10,000 most variable autosomal sites in FANS-isolated nuclei populations from adult cortex ($n$ = 212 donors). Using the adult PCA loadings, fetal samples were projected onto the same PCA space, showing a clear but attenuated difference between cell types in fetal cortex. Colors indicate the antibody labels used to isolate cortical nuclei. Colors are grouped into neuron-enriched (purple shades) and neuron-depleted (green shades) cell types.

(D) cg16424082, annotated to *METTL7A* on chromosome 12, is characterized by developmental-stage-specific differences in DNA methylation in neuron-enriched but not neuron-depleted nuclei populations.

(E) DNA methylation at cg04831806 annotated to *SDK1* on chromosome 7, a gene with a key role in neurodevelopment,[35] is characterized by a highly significant interaction between cell-type and developmental age (SATB2+: % change in DNA methylation per week = −4.25, $p$ = 6.76 × 10$^{-13}$).

*(legend continued on next page)*

component analysis (PCA) on the top 10,000 most variable autosomal sites calculated across the entire dataset highlighted that postnatal cell type explained the majority of variance in DNA methylation (principal component [PC]1, 71.5% of the variance), with developmental stage contributing significantly to both PC2 (17.1% of the variance) and PC3 (6.47% of the variance) (Figure 4A). We identified 6,531 DMPs between SATB2+ and SATB2− nuclei isolated from early/mid-fetal cortex (n = 37 donors, 8–20pcw). DNA methylation at these sites perfectly discriminated between NeuN+ (neuron-enriched) and NeuN− (neuron-depleted) nuclei in the postnatal cortex (Figure 4B) and with much larger effect sizes than in fetal cortex, highlighting that cell-specific DNA methylation profiles are established early in development prior to terminal differentiation. Compared to the differences identified between SATB2+ and SATB2− nuclei in fetal cortex, we identified more dramatic differences between NeuN+ and NeuN− nuclei in adult cortex (n = 212 donors, 18–108 years, 453,675 DMPs) (Table S10); although cell-type differences at these sites were significantly smaller in the fetal cortex (mean absolute effect size [%]: fetal = 1.61, adult = 14.4, t test $p < 1 \times 10^{-320}$), it is striking that they still discriminate between neuron-enriched and neuron-depleted nuclei populations during early fetal development (Figure 4C). Despite an overall positive correlation of DNA methylation differences between these nuclei populations in fetal and adult cortex across all sites tested (corr = 0.230; Figure S15A; Table S10), many cell-type DMPs were characterized by developmental-stage-specific effects (Figures S15B and S15C). Of particular interest were 1,427 sites that showed significantly different levels of DNA methylation between neuron-enriched and neuron-depleted nuclei in both fetal cortex and adult cortex but with an opposite direction of effect (Figure 4D). These results suggest that, while neuron-specific patterns of DNA methylation are established early in development, there are some notable shifts in patterns of cell-type-specific DNA methylation during development of the human cortex. Of note, these differences are predominantly driven by changes in DNA methylation in neuron-enriched and not in neuron-depleted cells.

### Derived cell-type proportions from DNA methylation data highlights cellular changes across cortex development

We used reference DNA methylation profiles from SATB2+, NeuN+, and NeuN− sorted nuclei obtained from late-fetal and early-postnatal human cortex, as well as embryonic stem cell (ESC) and neural progenitor cell (NPC) signatures previously described in Kim et al.[36] to deconvolute the proportion of each cell type in our bulk cortex prenatal dataset using the CETYGO package as previously described.[37] This analysis revealed a striking developmental increase in the proportion of SATB2+ cells across development (Figure S16). Concurrently, we

observe a decrease in the estimated proportion of ESC-like cells across this period, which is consistent with the loss of pluripotent cell types previously observed.[38] In contrast, estimated proportions of NeuN+ nuclei remain minimal throughout the early- and mid-fetal period, corroborating both our FANS data and transcriptomic analyses that demonstrate low NeuN expression during this developmental stage. These findings reinforce our conclusion that NeuN is not a reliable marker for identifying neurons in the early developing human cortex.

### Distinct trajectories of DNA methylation across cortex development between SATB2+ and SATB2− nuclei populations

We used a linear model, adjusting for sex and experimental batch, to characterize changes in DNA methylation in SATB2+ and SATB2− nuclei isolated from samples spanning early- to mid-fetal cortex development (n = 37 donors; Table S1). Effect sizes for bulk cortex dDMPs that were also tested in FANS-purified nuclei (n = 42,114) showed a high correlation with those in both SATB2+ (corr = 0.923) and SATB2− (corr = 0.870) samples (Figure S17). Although the power to detect experiment-wide significant developmental changes was limited within each nuclei population given the relatively small sample size, we identified 1,872 SATB2+ and 820 SATB2− dDMPs ($p < 9 \times 10^{-8}$) (Figure S18; Table S11). Of note, 4,088 of the dDMPs identified in the bulk cortex showed significant developmental changes in SATB2+ nuclei ($p < 1.19 \times 10^{-6}$, corrected for 42,114 dDMPs also tested in FANS-isolated nuclei), with 1,820 dDMPs showing significant developmental changes in SATB2− nuclei (Figures S19 and S20). Although there was a moderate correlation in developmental age effect sizes between the two cellular populations (corr = 0.564), many of the dDMPs identified in SATB2+ nuclei were distinct from those identified in SATB2− nuclei, in contrast to SATB2− dDMPs, which tended to be shared across both cell types (correlation between cell types: SATB2+ dDMPs = 0.634, SATB2− dDMPs = 0.873; Figure S21). There were some examples of cell-type-specific developmental effects; using an interaction model between age and cell type we identified 744 sites at which the developmental change in DNA methylation was significantly different between SATB2+ and SATB2− nuclei ($p < 9 \times 10^{-8}$) (Figure S22; Table S11). The most notable difference was at a site annotated to *SDK1* (cg04831806: interaction $p = 2.97 \times 10^{-19}$, SATB2+ effect = −4.25 [$p = 6.76 \times 10^{-13}$], SATB2− effect = −0.0984 [$p = 0.602$]) that encodes an immunoglobulin with a critical role in neuronal development and synapse formation[35] (Figure 4E), with other notable interaction effects seen for sites annotated to *SYT1*, *PEX14*, *OAT*, and *GJA1* (Figure S22). We used publicly available single-cell ATAC-seq data from multiple human fetal tissues[21] to test whether SATB2+-specific (n = 1,596) and SATB2−-specific (n = 548) autosomal dDMPs were enriched in

---

(F) Enrichment of SATB2+ dDMPs in cell-type-specific regions of open chromatin in the developing cortex identified by scATAC-seq.[21] SATB2+-specific autosomal dDMPs (n = 1,596) are most strongly enriched in peaks specific to excitatory neurons.

(G) Enrichment of SATB2− dDMPs in cell-type-specific regions of open chromatin in the developing cortex identified by scATAC-seq.[21] SATB2−-specific autosomal dDMPs (n = 548) are most strongly enriched in peaks specific to astrocytes. Color indicates direction of effect: red, significant enrichment; blue, significant depletion; gray, non-significant.

regions of open chromatin associated with the development of specific cell types. SATB2+-specific dDMPs were significantly and specifically enriched in excitatory neuron ATAC-seq peaks (effect size = 2.76, odds ratio = 15.7, corrected $p$ = $1.16 \times 10^{-86}$) with no such enrichment observed for dDMPs specific to SATB2− nuclei (effect size = −0.673, odds ratio = 0.510, corrected $p$ = 1). In contrast, dDMPs specific to SATB2− nuclei were enriched in peaks defining astrocytes only (effect size = 2.62, odds ratio = 13.7, corrected $p$ = $1.07 \times 10^{-40}$) (Figures 4F, 4G, and S23; Tables S12 and S13).

## Genes implicated in autism and schizophrenia are enriched for neurodevelopmental changes in DNA methylation

Autism and schizophrenia are highly heritable neurodevelopmental conditions hypothesized to arise from differences in early cortex development,[39] and recent exome- and whole-genome-sequencing studies have identified a number of rare but highly penetrant mutations associated with both conditions. We explored the extent to which DNA methylation sites annotated to both high-confidence autism genes (n = 233 "ategory 1" autism genes from the Simons Foundation Autism Research Initiative [SFARI] Gene database[40]) and schizophrenia genes identified from exome-sequencing (n = 32 genes identified by the SCHEMA consortium[41]) (Table S14) were enriched for dDMPs (see STAR Methods). Among autism and schizophrenia genes, there was a modest enrichment of genes associated with at least one bulk cortex dDMP compared to the background rate of 43.7% for all genes annotated to DNA methylation sites (SFARI autism and SCHEMA schizophrenia genes combined = 69.8% [$p$ = 0.0330], SFARI autism genes = 70.4% [$p$ = 0.0180], SCHEMA schizophrenia genes = 64.5% [$p$ = 0.119]; Figure S24; Table S15). The relative enrichment of dDMPs was even more striking in SATB2+ nuclei for both gene sets (background rate of genes associated with dDMPs in SATB2+ nuclei = 3.95%, SFARI autism and SCHEMA schizophrenia genes combined = 14.7% [$p$ = $6.61 \times 10^{-4}$], SFARI autism genes = 15.7% [$p$ = $8.74 \times 10^{-4}$], SCHEMA schizophrenia genes = 12.9% [$p$ = $2.38 \times 10^{-4}$]) (Figures S25A and S25B; Table S15). In contrast, only SCHEMA schizophrenia genes were significantly enriched for genes annotated to dDMPs in SATB2− nuclei (background rate of genes associated with dDMPs in SATB2− nuclei = 1.89%, SFARI autism and SCHEMA schizophrenia genes combined = 6.81% [$p$ = 0.155], SFARI autism genes = 6.01% [$p$ = 0.194], SCHEMA schizophrenia genes = 9.68% [$p$ = $2.67 \times 10^{-3}$]) (Figures S25C and S25D; Table S15). Given the range of effects among dDMPs, we next tested whether more significant dDMPs were more strongly enriched for autism and schizophrenia genes. We performed a tiered enrichment analysis across sequential association bins, starting with the top-ranked dDMPs and progressively including less strongly associated sites. Notably, we observed that the top-ranked dDMPs showed the strongest enrichment for genes associated with autism and schizophrenia (Figure S26).

Both autism and schizophrenia also have a large polygenic component, and we next examined the enrichment of dDMPs in genomic regions containing common disease variants identified by genome-wide association studies (GWASs) using MAGMA gene set analysis[42] with the most recent GWAS results for autism[43] and schizophrenia[44] (see STAR Methods). For schizophrenia, we observed an overall enrichment among bulk cortex dDMPs (effect size = 0.0496, $p$ = $7.43 \times 10^{-3}$). Cell-type-specific analyses suggest this association is driven by SATB2+ dDMPs (effect size = 0.0873, SE = 0.0446, $p$ = 0.0250) and not by SATB2− dDMPs (effect size = −0.0944, SE = 0.0613, $p$ = 0.938) (Figure S27). No significant MAGMA enrichment was observed for autism (Figure S27), likely reflecting the limited power of the current autism GWAS.

## DISCUSSION

We characterized changes in cortical DNA methylation during early- and mid-fetal development, exploring the extent to which these trajectories continued in the postnatal cortex and identifying differences in developmental patterns of DNA methylation between different cell types. We identified dramatic changes in DNA methylation associated with cortex development, with most of these effects specific to the prenatal period and distinct from age-associated changes in the postnatal cortex. We developed a novel FANS protocol to isolate SATB2+ nuclei from the developing cortex and identify trajectories of DNA methylation associated with the development of SATB2+ (neuron-enriched) and SATB2− (neuron-depleted) populations. Finally, we demonstrated an enrichment of developmentally dynamic DNA methylation sites annotated to genes implicated in autism and schizophrenia, supporting a role for neurodevelopmental processes in these conditions. This is, to our knowledge, the most extensive study of DNA methylation across the development of the human cortex and confirms the prenatal period as a time of considerable epigenomic plasticity in the human brain.

Several key findings emerge from our analyses. First, we show that the distribution of developmentally associated differentially methylated positions differs across genomic regions, being depleted in CGIs (likely due to their association with stably expressed housekeeping genes[45]) and enriched in regions of open chromatin identified in scATAC-seq analyses of fetal tissue development. Furthermore, although there was an overall enrichment of dDMPs becoming hypomethylated during development, in some genomic features, the converse was true; for example, ~90% of the dDMPs annotated to CGIs became hypermethylated over prenatal development. This supports observations from previous studies of changes in DNA methylation during early- and mid-gestation in brain and other tissues[13,15,25,26] and reflects the role of CGI methylation in mediating tissue-specific transcriptional programs during development.[46]

Second, the dramatic changes in DNA methylation observed in the developing cortex were largely specific to the prenatal period. These changes were distinct from the age-related changes seen in the postnatal cortex, and DNA methylation at dDMPs remained relatively stable after birth. Among the small subset (~2%) of dDMPs that did show significant postnatal changes in DNA methylation, many exhibited changes in the opposite direction to those seen across development of the fetal cortex. Specifically, sites that were hypomethylated during the prenatal period often became significantly hypermethylated

postnatally, which is consistent with previous findings.[15] These results confirm that the prenatal period is a time of intense epigenetic activity in the developing cortex, with more extensive and larger DNA methylation changes compared to the postnatal period.

Third, our findings demonstrate that DNA methylation changes during cortex development are not uniformly linear. Using a nonparametric Gaussian process regression model, we identified thousands of sites whose DNA methylation changed nonlinearly during fetal cortex development. These nonlinear changes could provide important insights into epigenetic milestones during key stages of brain development, many of which could be missed by standard linear regression approaches. Clustering these sites into six distinct co-methylation modules revealed shared nonlinear trajectories and highlighted clear switch points in DNA methylation during development, particularly between 12 and 15pcw. For example, nonlinear sites in the blue module were characterized by a rapid initial decrease in DNA methylation, plateauing around 15 pcw, and pathway analysis of these sites revealed terms relating to neuron projection and synaptic signaling. Additionally, sites in the turquoise module exhibited a rapid increase in DNA methylation until 12 pcw. These sites were enriched for extracellular matrix and neural cell migration pathways, which are believed to peak around 12 pcw.[47,48] Collectively, these findings highlight the utility of using nonlinear models to identify distinct patterns of DNA methylation in the developing fetal cortex that may be indicative of developmental switch points that would not have been detected using a standard linear model.

Fourth, we find that there are distinct patterns of DNA methylation between different cell-type populations. The period of cortical development is marked by rapid neurogenesis and the terminal differentiation of excitatory neurons, which arise from radial glial progenitor cells in the ventricular and subventricular zones, with this process starting ~7 pcw and peaking between 12 and 20 pcw.[47,49] To explore DNA methylation changes accompanying neurogenesis, we optimized a method to isolate neuron-enriched and neuron-depleted nuclei populations, identifying SATB2 as a robust marker of prenatal neurons. Our analysis revealed that cell-type differences accounted for a large proportion of the variation in DNA methylation, with consistent differences between neuron-enriched and neuron-depleted cells in both fetal and adult samples. Our analyses highlighted cell-type-specific DNA methylation trajectories during cortex development, with changes in bulk cortex primarily reflecting those occurring in developing neurons. Of note, analysis of a fetal scATAC-seq dataset revealed an enrichment of SATB2+-specific dDMPs in regions of open chromatin associated with the development of excitatory neurons and an enrichment of SATB2−-specific dDMPs in regions of open chromatin associated with astrocytes. These results confirm the interplay between DNA methylation and chromatin organization during development[50] and suggest that dynamic changes in DNA methylation play an important role in establishing the broader cell-type-specific epigenetic landscape of the developing human cortex.

Finally, we found evidence that dDMPs are enriched in sites annotated to genes associated with autism and schizophrenia, two highly heritable neurodevelopmental conditions that have

been hypothesized to arise from differences in early cortex development.[39,51,52] Genes robustly associated with both schizophrenia (from the SCHEMA consortium[41]) and autism (from the SFARI gene database[40]) were enriched for dDMPs, providing further evidence to support the notion that altered gene regulation during fetal brain development is involved in these conditions. dDMPs were also enriched in the vicinity of common variants associated with schizophrenia by GWAS, although no enrichment was observed for common variants associated with autism, possibly reflecting the low power of the current autism GWAS. Our data support the hypothesis that a significant proportion of the genes harboring variants that confer risk for neurodevelopmental conditions have regulatory effects that manifest during the development of the human cortex.

## Conclusions

Our study uncovers highly dynamic and widespread changes in DNA methylation across the human cortex during development. These dramatic temporal shifts in DNA methylation are predominantly confined to the prenatal period, differing significantly from the more gradual, age-related changes observed in the postnatal cortex. Importantly, the enrichment of dDMPs annotated to genes associated with schizophrenia and autism offers novel insights into the developmental mechanisms underlying these conditions. These findings highlight how early epigenetic modifications may mediate the onset of neurodevelopmental phenotypes, paving the way for future mechanistic research.

## Limitations of the study

This study has several limitations that should be taken into consideration. First, legal restrictions on later-term abortions limited our access to cortical tissue from more advanced stages of fetal development. However, the relative stability of DNA methylation at dDMPs between the oldest fetal samples and adult cortex samples (Figure 3G) suggests that the magnitude of epigenetic changes during late pregnancy and early postnatal life is much smaller than those observed during early fetal development. Of note, the age range of samples profiled in our study (6 pcw to 108 years) is larger than of those profiled in previous analyses of brain DNA methylation across the life course such as that reported by Numata et al.[15] (14 pcw to 84 years). Given the dramatic changes seen across our earliest fetal samples, the extended range of fetal cortex ages is a particular strength of the current study. Second, while the Illumina EPIC array enables precise quantification of DNA methylation at single-base resolution for sites annotated to the vast majority of genes and CGIs and is more comprehensive than assays used in previous studies,[13,15] it covers only a small proportion of CpG sites in the human genome, and these are not evenly distributed across all genomic features. As sequencing costs decline, future research should utilize sequencing-based technologies to comprehensively profile the epigenome across cortical development with larger sample sizes. Third, our ability to explore non-CpG (i.e., CH) methylation was constrained by the very small number of non-CG sites included on the Illumina EPIC array ($n = 1,277$; 0.158% of the sites included in our final dataset); this is potentially important given increasing evidence for a role

of CH methylation in the brain.[14] Only one of these sites is included in the list of 50,913 bulk cortex dDMPs (ch.14.30061788F [not annotated to any gene], chromosome 14, $p = 4.39 \times 10^{-9}$), although future studies using sequencing-based approaches are required to systematically explore non-CpG methylation. Fourth, our prenatal and postnatal cortex samples were processed separately, potentially confounding our comparisons between them. Of note, however, the higher variance at dDMPs among prenatal cortex samples compared to postnatal cortex samples (Figure 3A) is specific to dDMPs, and the magnitude of DNA methylation differences between early prenatal samples and postnatal samples (near 100% in many instances) is unlikely to reflect batch effects. Fifth, despite this being the largest study of cell-type-specific DNA methylation changes in human cortex development to date, the labor-intensive nature of isolating purified nuclei via FANS limited us to collecting SATB2+ and SATB2− fractions from only a subset of donors, which reduced our power to detect dDMPs in these populations. Nevertheless, we observed strong concordance with bulk cortex data and clear evidence of cell-type-specific developmental trajectories. Importantly, sorting only for SATB2 does not enable us to capture the full extent of cellular diversity in the fetal cortex, and, because the SATB2− population was not positively sorted, it contained a mix of different cell types and is more heterogeneous than the SATB2+ population, which is relatively pure. For example, some neuronal nuclei—particularly derived from inhibitory neurons—would be expected in the SATB2− population, as SATB2 is not robustly expressed in these cells. Sixth, our study could not distinguish between DNA methylation and its oxidized form, DNA hydroxymethylation, due to limitations of sodium bisulfite-based approaches.[53] This distinction is potentially important, as DNA hydroxymethylation is abundant in the central nervous system and known to impact gene expression.[14] Future studies should aim to differentiate these modifications using techniques such as nanopore sequencing.[54] Lastly, we did not obtain gene expression data from these samples, preventing direct conclusions about the transcriptional impact of the observed DNA methylation changes. However, by integrating our dDMPs with scATAC-seq peaks from fetal brain,[21] we were able to show that the DNA methylation changes observed reflect broader shifts in gene regulation throughout cortical development.

## RESOURCE AVAILABILITY

### Lead contact
Requests for further information and resources should be directed to and will be fulfilled by the lead contact, Jonathan Mill (j.mill@exeter.ac.uk).

### Materials availability
This study did not generate new unique reagents.

### Data and code availability
- DNA methylation data generated as part of this study are publicly available via Gene Expression Omnibus (GEO) accession numbers GSE289191 (bulk fetal and child cortex) and GSE289184 (FANS sorted fetal and child cortex).
- Bulk adult cortex data are available at GSE197305,[31] and sorted adult data are available at GSE279509.[55]
- Code relating to the analyses reported here can be found on GitHub at https://github.com/alicemfr/DevCortexDNAm/ (https://doi.org/10.5281/zenodo.16876128).

## CONSORTIA

The members of the APEX Consortium are Simon Baron-Cohen, Carrie Allison, Varun Warrier, Alex Tsompanidis, Deep Adhya, Rosie Holt, Joanna Davis, Genie Gu, Yira Zhang, Niran Okewole, Omar Al-Rubaie, Daniel H. Geschwind, Ramin Ali Marandi Ghoddousi, Alexander E.P. Heazell, Jonathan Mill, Alice Franklin, Rosemary Bamford, Matthew E. Hurles, Hilary C. Martin, Mahmoud Mousa, David H. Rowitch, Kathy K. Niakan, Graham J. Burton, Fateneh Ghafari, Deepak P. Srivastava, Lucia Dutan-Polit, Adam Pavlinek, Laura Sichlinger, Roland Nagy, Madeline A. Lancaster, Jose Gonzalez-Martinez, Tal Biron-Shental, Lidia V. Gabis, Dori Floris, Richard Bethlehem, Mike Lombardo, Marcin Radecki, Meng-Chuan Lai, Yeshaya David Greenberg, Elizabeth Weir, Florina Uzefovsky, Yumnah Khan, Juan Pablo Del Rio, Anna Penn, Rebecca Knickmeyer, and Kyriaki Kosidou.

The members of the Youth-GEMs Consortium are Bart Rutten, Sinan Gülöksüz, Therese van Amelsvoort, Lotta-Katrin Pries, Bochao Danae Lin, Angelo Arias-Magnasco, Erika van Hell, Mary Cannon, David Cotter, Melanie Föcking, Subash Raj Susai, Elisabeth Binder, Jim van Os, Jeroen Pasterkamp, Anna Wiersema, Marco Boks, Winni Schalkwijk, Karim Lekadir, Esmeralda Ruiz Pujadas, Noussair Lazrak, Ian Kelleher, Jenni Leppänen, Valentina Kieseppä, Simona Karbouniaris, Lisette van der Poel, Mariël Kanne, Marijke Kolk, Hanske Douwenga, Jordi Rambla, Arcadi Navarro, Liina Nagirnaja, Aldar Cabrelles Munoz, Lauren A. Fromont, Jaanus Harro, Triin Kurrikoff, Reigo Reppo, Dejan Stevanovic, Aleksa Milevic, Jasna Jancic, Marija Nikolic, Maria Bulgheroni, Laura Giani, Margherita La Gamba, Tomislav Franic, Mia Plenkovic, Covadonga M. Diaz-Caneja, Celso Arango, Marta Ferrer-Quintero, Renzo Abregú-Crespo, Emily Guerra-Blacio, Nuria Martín-Martínez, Christel Middeldorp, Enda M. Byrne, Swathi Hassan Gangaraju, Sushma Marla, Jonathan Mill, Eilis Hannon, Emma Dempster, Philippa Wells, Robin Murray, Andrea Danese, Alexander L. Richards, Lucy Riglin, and Michael C. O'Donovan.

## ACKNOWLEDGMENTS

This work was supported by grants from the Simons Foundation for Autism Research (SFARI) (grant number 573312 awarded to J.M. and grant number 809383 awarded to the APEX consortium), grants from the UK Medical Research Council (grant MR/R005176/1 awarded to J.M. and grant MR/W017156/1 awarded to N.E.C.), and the European Union's Horizon Europe program (YOUTH-GEMs, grant agreement no. 101057182). The human prenatal material was provided by the Human Developmental Biology Resource (funded by Medical Research Council [MRC]/Wellcome Trust grant [099175/Z/12/Z]; https://www.hdbr.org). Sequencing infrastructure was supported by a Wellcome Trust Multi User Equipment Award (WT101650MA awarded to J.M.) and MRC Clinical Infrastructure Funding (MR/M008924/1 awarded to J.M.). This study was also supported by the National Institute for Health and Care Research Exeter Biomedical Research Centre. The views expressed are those of the author(s) and not necessarily those of the NIHR or the Department of Health and Social Care. Components of the graphical abstract were created with BioRender.com. We would like to thank Andreas Adinatha and Eliora V. Buyamin for their help with snRNA-seq analyses.

## CellPress

## AUTHOR CONTRIBUTIONS

Conceptualization, J.M., E.H., and E.D.; data curation, A.F.; formal analysis, A.F., E.M.W., M.F., and N.E.C.; funding acquisition, J.M.; investigation, J.P.D., G.E.T.B., R.B., B.C., and J.B.; methodology, A.F., N.O., and E.H.; project administration, J.M.; supervision, J.M. and E.H.; visualization, A.F.; writing – original draft, A.F. and J.M.; writing – review & editing, all authors.

## DECLARATION OF INTERESTS

The authors declare no competing interests.

## STAR★METHODS

Detailed methods are provided in the online version of this paper and include the following:

- ● KEY RESOURCES TABLE
- ● EXPERIMENTAL MODEL AND STUDY PARTICIPANT DETAILS
- ● METHOD DETAILS
  - ○ DNA methylation profiling
  - ○ FANS isolation of specific nuclei populations
  - ○ Single nuclei RNA-Seq
- ● QUANTIFICATION AND STATISTICAL ANALYSIS
  - ○ Whole cortex life-course DNA methylation dataset
  - ○ Estimating biological age with epigenetic clocks
  - ○ Cell-type-specific life-course DNA methylation dataset
  - ○ Development-associated DNA methylation changes
  - ○ Nonlinear trajectories during cortex development
  - ○ snRNA-seq analysis
  - ○ Cell-type-specific life-course trajectories
  - ○ Genomic feature and chromosome enrichment
  - ○ scATAC-seq peak enrichment
  - ○ Prenatal cortex cell-type deconvolution
  - ○ Enrichment of dDMP-annotated genes among autism and schizophrenia genes

## SUPPLEMENTAL INFORMATION

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

## STAR★METHODS

### KEY RESOURCES TABLE

| REAGENT or RESOURCE | SOURCE | IDENTIFIER |
|---|---|---|
| **Antibodies** | | |
| Hoechst 33342 | Abcam | #ab228551 |
| Alexa Fluor® 488 Anti-SATB2 antibody [EPNCIR130A] | Abcam | #ab196316; RRID:AB_2943096 |
| **Biological samples** | | |
| Human cortex tissue (Fetal; Child) | Human Developmental Biology Resource (HDBR) & MRC UK Brain Bank network | www.hdbr.org; https://ukbbn.brainsfordementiaresearch.org/public/en/ |
| **Critical commercial assays** | | |
| Chromium Next GEM Single Cell 3' Reagent Kits v3.1 | 10x Genomics | PN-1000121/PN-1000120/PN-1000213 |
| Illumina Infinium MethylationEPIC BeadChip | Illumina | WG-317-1003 |
| **Deposited data** | | |
| Human Cortex DNA Methylation (Bulk): Fetal (6-33pcw); Child (0-8years) | This paper | GSE289191 |
| Human Cortex DNA Methylation (FANS): Fetal (8-28pcw); Child (0-8years) | This paper | GSE289184 |
| Human Cortex DNA Methylation (Bulk): Adult (25-104years) | Shireby et al.[31] | GSE197305 |
| Human Cortex DNA Methylation (FANS): Adult (18-108years) | Walker et al.[55] | GSE279509 |
| BrainSeq Consortium Human DLPFC RNA-seq (Fetal; Child; Adult) | Jaffe et al.[33] | https://eqtl.brainseq.org/phase1/ |
| **Software and algorithms** | | |
| dmrff | Suderman et al.[19] | https://github.com/perishky/dmrff |
| FetalClock | Steg et al.[16] | https://github.com/LSteg/EpigeneticFetalClock |
| agep within the wateRmelon package | Horvath[27] | https://bioconductor.org/packages/wateRmelon/ |
| CorticalClock | Shireby et al.[28] | https://github.com/gemmashireby/CorticalClock |
| missMethyl | Phipson et al.[56] | https://bioconductor.org/packages/missMethyl/ |
| CETYGO | Vellame et al.[37] | https://github.com/ejh243/CETYGO |
| AnnotationHub | Morgan and Shepherd[57] | https://bioconductor.org/packages/AnnotationHub/ |
| MAGMA | de Leeuw et al.[42] | https://cncr.nl/research/magma/ |
| GPMethylation | This paper | https://github.com/owensnick/GPMethylation.jl |
| Illumina Infinium MethylationEPIC v1.0 B4 Manifest File | Illumina | https://emea.support.illumina.com/downloads/infinium-methylationepic-v1-0-product-files.html |
| WGCNA | Langfelder & Horvath[23] | https://CRAN.R-project.org/package=WGCNA |
| CellRanger | 10x Genomics | https://www.10xgenomics.com/support/software/cell-ranger/downloads |
| Seurat | Stuart et al.[58]; Hao et al.[59] | https://github.com/satijalab/seurat |
| DoubletFinder | McGinnis et al.[60] | https://github.com/chris-mcginnis-ucsf/DoubletFinder |
| scCustomize | Marsh[61] | https://github.com/samuel-marsh/scCustomize/ |
| **Other** | | |
| Supporting code | This paper | https://github.com/alicemfr/DevCortexDNAm; https://doi.org/10.5281/zenodo.16876128 |
| Human Fetal Cell-Type-Specific ATAC-Seq Peaks | Domcke et al.[21] | N/A |

*(Continued on next page)*

| *Continued* | | |
|---|---|---|
| REAGENT or RESOURCE | SOURCE | IDENTIFIER |
| Autism GWAS Summary Statistics | Grove et al.,[43] Psychiatric Genomics Consortium | iPSYCH-PGC_ASD_Nov2017; https://figshare.com/articles/dataset/asd2019/14671989 |
| Schizophrenia GWAS Summary Statistics | Trubetskoy et al.,[44] Psychiatric Genomics Consortium | PGC3_SCZ_wave3.primary.autosome.public.v3; https://figshare.com/articles/dataset/scz2022/19426775 |

## EXPERIMENTAL MODEL AND STUDY PARTICIPANT DETAILS

Cortex tissue from prenatal and childhood donors was acquired from the Human Developmental Biology Resource (HDBR) (http://www.hdbr.org) and the MRC UK Brain Banks network (https://ukhealthdata.org/members/mrc-uk-brain-banks-network). Ethical approval for the HDBR was granted by the Royal Free Hospital research ethics committee under ref. 08/H0712/34 and Human Tissue Authority (HTA) material storage license 12220; ethical approval for the MRC Brain Bank was granted under ref. 08/MRE09/38. The age of donors was determined by Carnegie staging in the case of embryonic samples and foot and knee to heel length measurements for fetal samples. Apart from sex, no additional phenotypic or demographic information was available. A full overview of the samples used in this study is provided in Table S1.

## METHOD DETAILS

### DNA methylation profiling

Genomic DNA was isolated from each tissue sample using a standard phenol-chloroform extraction protocol and assessed for quality and purity using spectrophotometry. DNA methylation was quantified using the Illumina Infinium HumanMethylation EPIC array (Illumina Inc), which interrogates >850,000 DNA methylation sites across the genome. Briefly, the EZ-96 DNA Methylation-Gold kit (Zymo Research) was used for sodium bisulfite conversion prior to the quantification of DNA methylation using the Illumina EPIC array. All subsequent statistical analyses were performed in R version 3.6.0 unless otherwise stated. Raw Illumina EPIC data was processed using the *wateRmelon* R package as previously described.[62] Briefly, DNA methylation data were loaded from raw IDAT files and processed through a standard quality control pipeline that includes the following steps: 1) Checking methylated and unmethylated intensities and excluding samples where this was <500; 2) Calculating bisulfite conversion efficiency and excluding samples with median <80%; 3) Principal component analysis of the X and Y chromosomes to confirm reported sex; 4) Confirming sample (un)relatedness using the 59 'rs' SNP probes on the EPIC array; samples with correlation >80% to another unrelated sample were excluded; 5) Detection of outlier samples with the *outlyx* function; 6) Using the *pfilter* function to identify and exclude samples with >1% of probes with detection $p > 0.05$ and probes where >1% of samples had a detection $p > 0.05$. 7) Exclusion of probes previously identified as cross-hybridising or polymorphic.[63] 8) Normalisation with *dasen*. Data generated on the purified nuclei populations were confirmed through Principal Component Analysis (PCA) to cluster within cell-type and were normalised within each cell-type separately. EPIC array probes were annotated to genes according to the Illumina Infinium MethylationEPIC v1.0 B4 Manifest File (available at: https://emea.support.illumina.com/downloads/infinium-methylationepic-v1-0-product-files.html).

### FANS isolation of specific nuclei populations

Neuron-enriched and neuron-depleted nuclei fractions were isolated from 38 fetal cortex tissue samples using a fluorescence-activated nuclei sorting (FANS) protocol optimized by our group. Briefly, following tissue homogenization and nuclei purification using sucrose gradient centrifugation we used a FACS Aria III cell sorter (BD Biosciences) to simultaneously collect populations of SATB2+ (AbCam, Cat No: ab196316, dilution: 1:1000) and SATB2- immunolabeled populations from fetal cortex prior to DNA methylation profiling. For each nuclei population, ~200,000 nuclei were collected for extraction of genomic DNA using a standard phenol: chloroform extraction protocol and DNA methylation was profiled using the Illumina EPIC array as described above. Our protocol detailing our SATB2+ FANS protocol is available on protocols.io[64] as a resource to the community.

### Single nuclei RNA-Seq

50mg of cortex tissue was homogenized using the recommended protocol from 10X Genomics (CG000124, Rev F) with a few minor amendments. Briefly, ribolock RNase-inhibitor (Thermo Scientific, EO0382) was added to the lysis buffer (0.4 U/ul) and staining buffer (0.2U/ul). Following antibody staining, nuclei were collected by FANS as described above. Nuclei were then spun and resuspended in staining buffer. Nuclei suspensions were assessed for the presence of debris and manually counted on a haemocytometer before proceeding with single-nucleus capture using the 10x Genomics Single-Cell 3′ technology. Targeted recovery of 3000 nuclei per sample was used. The 10x Chromium Next GEM Single Cell 3′ protocol (v3.1) was followed for expression library preparation from single nuclei. cDNA and final library quantification, quality control and fragment size determination were performed using the

D5000 high sensitivity ScreenTape assay and reagents (Agilent technologies). Sequencing libraries were pooled and sequenced on an Illumina NovaSeq6000 sequencer.

## QUANTIFICATION AND STATISTICAL ANALYSIS

### Whole cortex life-course DNA methylation dataset

After quality control and normalization, the fetal cortex DNA methylation dataset ($n$ = 91, age range = 6–23 pcw) included 807,806 DNA methylation sites (789,981 autosomal). To generate a life-course dataset we incorporated DNA methylation data from 16 late-gestation fetal and child donors (age range = 26 pcw - 8 years) and 661 adult donors (age range = 25–104 years) using data previously reported by our group (Gene Expression Omnibus (GEO) accession number GSE197305[31]). The final bulk cortex life-course dataset included DNA methylation data from 768 donors (Table S1) and included data for 41,518 of the 50,913 dDMPs identified in the fetal cortex.

### Estimating biological age with epigenetic clocks

Age estimates for bulk fetal cortex samples were calculated using the *FetalClock* function[16] available at https://github.com/LSteg/EpigeneticFetalClock. Age estimates for postnatal bulk cortex samples used the *agep* function from the wateRmelon R package[27] in addition to the *CorticalClock* function[28] available at https://github.com/gemmashireby/CorticalClock.

### Cell-type-specific life-course DNA methylation dataset

We combined normalized neuron-enriched (SATB2+) and neuron-depleted (SATB2-) DNA methylation data derived from 37 early-/mid-fetal donors (SATB2+ $n$ = 34, SATB2- $n$ = 33, age range = 8–20 pcw), 1 late-fetal donor (SATB2+ $n$ = 1, SATB2- $n$ = 1, age = 28 pcw) and 9 child donors (SATB2+ $n$ = 9, SATB2- $n$ = 9, age range = 0–8 years), generated using the FANS protocol described above, with data from 212 adult donors (neuron-enriched (NeuN+) $n$ = 178, neuron-depleted (SOX10+, IRF8+ and SOX10-/NeuN-) $n$ = 335, age range = 18–108 years) generated previously by our group as part of a large-scale study quantifying DNA methylation in human purified cortical nuclei.[55] The final life-course dataset contained DNA methylation data for FANS-isolated nuclei populations from 259 donors (Table S1) across 693,964 shared sites (679,917 autosomal).

### Development-associated DNA methylation changes

We used a multiple linear regression model to test the association between developmental age in post-conception weeks (pcw) and DNA methylation, whilst controlling for sex and experimental batch. A site was considered to be a significant dDMP if its $p$ value surpassed an experiment-wide significance threshold of $p < 9 \times 10^{-8}$, previously determined to adequately control for the false positive rate of DNA methylation studies on the EPIC array.[17] Regional analysis of dDMPs was performed using the *dmrff* R package to identify differentially methylated regions (dDMRs). dDMRs were defined as regions with three or more dDMPs within a maximum window of 500 bp and reaching a Bonferroni-adjusted $p < 0.05$. dDMRs were annotated to genes using the *AnnotationHub* R package[57] with hg19 genome build.

### Nonlinear trajectories during cortex development

To determine sites with nonlinear trajectories, we first applied additional filtering. A leave-one-out $Z$ score approach was used to detect and remove the contribution of outlier samples (those with Z-scores >5 standard deviations from the mean). Non-variable DNA methylation sites (sites whose range of the middle 80% of values is <5% DNA methylation), and constant sites (sites with DNA methylation consistently >90% or <10% for all samples), were also removed. Using https://github.com/owensnick/GPMethylation.jl in Julia v1.6.1, we performed exact Gaussian process regression and model selection to classify DNA methylation sites as either constant, linear or nonlinear (Matern 5/2 kernel). Hyperparameters were selected by optimization of the marginal likelihood. Classification was determined by considering the kernel that provided the greatest optimized marginal log likelihood. Nonlinear sites ($n$ = 175,419) were further refined by considering the log likelihood ratio (LLR) of the nonlinear kernel against both the constant and linear kernels (Figure S28). The timescale of a DNA methylation site (traditionally called length-scale) relates to the periodicity of the oscillation, measured in post-conception weeks (pcw). To determine the period of one complete oscillation, the timescale can be multiplied by $2\pi$ sqrt($^3/_5$) ($\approx$5), i.e., a timescale of 1 equates to a complete oscillation every $\approx$5 pcw.[65,66] To focus on sites with strong evidence for nonlinear behavior with biologically meaningful timescales, sites with LLR <2 and timescale <10 were excluded, leaving 73,638 high-confidence nonlinear sites with timescales between 10.0 and 106 (Figure S28). We used the weighted gene correlation network analysis (*WGCNA*) R package[23] to identify modules of co-methylated nonlinear sites. Using the *blockwiseModules* function, a signed network was constructed using a soft threshold of 12, as recommended for this sample size.[67] The "gray" module was excluded from further analysis due to this being a non-specific module. Representative module plots (Figure 2) were constructed by calculating the first principal component (eigengene) of each module. "Hub sites" were defined as sites whose DNA methylation pattern most highly correlated with the module eigengene (Figure S29). Pathway analysis was performed using the *missMethyl* R package[56] on nonlinear sites within each module separately, with the background set as all other array sites.

### snRNA-seq analysis

Library demultiplexing, generation of FASTQ files, alignment of reads and quantification of unique molecular identifiers (UMIs) was performed using CellRanger software with default parameters (10x Genomics, v6.0.2). A pre-mRNA reference file (GRCh38) was used to ensure unspliced intronic reads were captured. snRNA-seq data was analyzed using the *Seurat* package[58,59] in R (v4.1 and 3.6). Doublet identification was performed using the *DoubletFinder* R package.[60] Cells were filtered prior to clustering on the basis of genes per cell (nFeature_RNA, data set specific) and percentage mitochondrial reads per cell (greater than 5% mitochondrial reads excluded). Data was log normalized and highly variable genes were selected. The data was then scaled (linear transformation) and variation owing to mitochondrial gene expression was regressed out. PCA was performed followed by clustering using a KNN graph. Dimensionality reduction was performed with t-distributed Stochastic Neighbor Embedding (tSNE) or uniform Manifold Approximation and Projection (uMAP). Clusters were manually annotated using analysis of differentially expressed marker genes from previous scRNA-seq studies[68–70] and plots were generated using the *scCustomize* package.[61]

### Cell-type-specific life-course trajectories

Linear mixed-effects models were applied to test cell-type (neuron-enriched vs. neuron-depleted) associations across the 579,541 DNA methylation sites shared between fetal and adult nuclei-sorted samples. Fixed effects included age (prenatal age in post-conception weeks; postnatal age in years), sex and array plate number and individual ID was included as a random effect to account for multiple measurements from each individual. SOX10+ and IRF8+ dummy variables were included as fixed effect covariates within the adult model to account for differences between the cell-type sub-populations within neuron-depleted nuclei (SOX10+, IRF8+ and SOX10-/NeuN-). An ANOVA was used to determine probe significance by whether the addition of the variable of interest (cell-type) significantly improved the model ($p < 9 \times 10^{-8}$). Downstream analyses used the mixed-effects model coefficients.

### Genomic feature and chromosome enrichment

DNA methylation sites were annotated to genomic and CpG features according to the Illumina EPIC manifest. Enrichment of DMPs within features and chromosomes was tested with a chi-squared test. To determine if there was an overall enrichment of DMPs for a specific direction of effect (hypomethylated or hypermethylated), a one-tailed binomial test was used with probability of success equal to the proportion across all sites.

### scATAC-seq peak enrichment

We utilized publicly available scATAC-seq data generated by Domcke et al.[21] to interrogate the top 10,000 most specific chromatin peaks for 54 fetal human cell-types (available at GEO accession number GSE149683, Table S16). For each cell-type, autosomal DNA methylation sites were annotated with a binary variable indicating whether they resided within the genomic coordinates of each peak. We used logistic regression to test the association between a DNA methylation site being a DMP and being in a peak, whilst controlling for all other cell-type peaks.

### Prenatal cortex cell-type deconvolution

ESC ('Embryonic stem (ES) cells H9') and EPC-derived NPC ('ES-derived neural precursor cells') DNA methylation reference data were downloaded from GEO (accession number GSE38214) and combined with late-fetal (20–28pcw) and early postnatal (0 – 8years) SATB2+ samples (this study) in addition to postnatal (18–108years) NeuN+ and NeuN- samples (this study). The *pickCompProbesMatrix* function of the CETYGO R package[37] was used with numProbes = 100 and probeSelect = "any" to select DNA methylation sites that significantly distinguish the input cell-types, regardless of direction of effect, which are used as training data in the deconvolution model.

### Enrichment of dDMP-annotated genes among autism and schizophrenia genes

To test whether dDMPs were significantly enriched for gene sets associated with autism and schizophrenia,[40,41] we first annotated DNA methylation sites to genes as previously described. At the gene-level, we used logistic regression to test the relationship between DMP status (a binary variable of whether a gene contains at least one DMP) and gene set membership (whether the gene belongs to the gene set), whilst controlling for gene size (number of DNA methylation sites annotated to the gene). We further tested the extent to which the significance of the DMP impacts the association with each gene set. DMPs were ranked most to least significant, and the enrichment test was performed on bins of DMPs of increasing size. For each iteration, DMPs not part of the current bin were excluded from the analysis, such that the background set remained consistent across tests. Enrichment analyses with respect to common genetic variants associated with autism[43] and schizophrenia[44] were performed using MAGMA v1.10.[42] Non-variable DNA methylation sites and sites not annotated to a gene were excluded prior to analysis. GWAS-associated SNPs were subset to common variants (minor allele frequency (MAF) $\geq$ 1% and imputation information score >0.8) and SNP-associated GWAS *p* values were combined into gene-level *p* values using a window of 35 kb upstream and 10 kb downstream of the gene, as recommended.[71] One-tailed MAGMA gene set analysis tested the association between gene-level SNP *p* values and DMP status, whilst controlling for gene size (number of DNA methylation sites annotated to the gene).

