## [Document S2. Transparent peer review records for Franklin et al. · Cell Genomics]

Cell-type-specific DNA methylation dynamics in the prenatal and postnatal human cortex

Alice Franklin, Jonathan P. Davies, Nicholas E Clifton, Georgina E T Blake, Rosemary Bamford, Emma M Walker, Barry Chioza, Martyn Frith, APEX Consortium, Youth-GEMs Consortium, Joe Burrage, Nick Owens, Shyam Prabhakar, Emma Dempster, Eilis Hannon, Jonathan Mill

Summary

Initial submission: Received : Feb 21, 2025

Scientific editor: Sara Rohban

First round of review: Number of reviewers: 2
Revision invited : Mar 27, 2025
Revision received : Jul 08, 2025

Second round of review: Number of reviewers: 1
Revision invited : Jul 28, 2025
Revision received : Aug 08, 2025

Third round of review: Number of reviewers: 1
Accepted : Aug 27, 2025

Data freely available: YES

Code freely available: YES

This transparent peer review record is not systematically proofread, type-set, or edited. Special characters, formatting, and equations may fail to render properly. Standard procedural text within the editor's letters has been deleted for the sake of brevity, but all official correspondence specific to the manuscript has been preserved.

Referees' reports, first round of review

Reviewer #1:

This paper by Franklin et al analyzes DNA methylation profiles of the human cerebral cortex throughout prenatal and postnatal development, as well as aging, covering ages from the first trimester to late adulthood (>100 years of age). They utilize methylation arrays to perform bulk DNA methylation profiling and describe hyper and hypomethylated changes associated with different stages of development. This paper is an important resource and provides some incremental insights into our understanding of human brain development. However, there are several potential issues I believe authors should address to improve the paper:

1. Similar analysis has been done before more than 10 years ago in the Numata et al Am J Hum Genet 2012 paper. This paper used a cohort of samples with a similar age distribution and size (actually, a bit larger). Therefore, it would be important to explain how the current paper improves on this previous studies (e.g. by analyzing additional developmental stages, using improved method etc).
2. Similarly, it would be important to clearly highlight the findings that are novel and were not shown by the Numata paper.
3. If it is feasible, performing a meta-analysis of the Numata data with the current dataset would greatly improve the paper.
4. Is there a way to analyze CG and CH methylation using the data the authors produced? Given the importance and enrichment of CH methylation in neurons during postnatal development, this would be an important analysis.
5. SATB2 sorting is used to enrich for neurons. However, SATB2 is enriched in upper-layer projection neurons in the cortex. Authors need to show using qRT-PCR and/or immunostaining what kind of population they are actually enriching.
6. It is not clear whether all p values were corrected for multiple comparisons. This needs to be clearly indicated everywhere (corrected or uncorrected p value).

Reviewer #2:

Summary: Franklin et al generated DNA methylation (DNAm) data on the Illumina EPIC microarray across pre- (n=91 early-mid and n=4 mid-late) and post-natal (n=671) cortical development. The authors identified extensive age associations in pre-natal development, including linear/additive and more complex trajectories of DNAm levels that were more likely to occur in transcriptionally-relevant genomic annotations (open chromatic regions, CGI shores). The authors compared these changes to postnatal DNAm changes and found little overlap in directionality and magnitude. Due to extensive cellular composition changes occurring over cortical development, the authors developed and applied a SATB2 nuclei sorting strategy and characterized cell type-enriched methylation across development and aging. Lastly, the authors identified overlap between developmental changes in DNAm levels and genetic associations to autism spectrum disorder and schizophrenia. Overall, the paper was well written and easy to follow, particularly among the bulk analyses. My main concern was around the interpretation of the SATB2 nuclei sorting, and what cell type(s) were actually be compared in the resulting SATB2- and SATB2+ DNAm profiles with lesser potential concerns around how pre- and post-natal samples were jointly normalized to make samples directly comparable for analysis.

Comments:

- Prenatal age associations: these analyses are well described and easy to follow. However, I would probably integrate the non-linear and WCGNA modeling into the first results subsection describing age-related changes during pre-natal life since they generally support the extensive DNAm changes occurring during this critical period. Then the authors should perform analogous enrichment analyses as the linear changes (to more active genomic states) for CpGs with non-linear changes and with various WCGNA clusters to help better interpret/contextualize them.

- Pre- versus post-natal age associations:
 - o It doesn't look like the pre- and post-natal DNAm samples were generated in the same batches, which might limit direct comparisons of variance (eg Figure 2A). Can the authors clarify if these samples were at least processed+normalized together with dasen for these analyses? Additional processing techniques, like normalizing using the negative control probes across the arrays, could also help reduce variance due to different batches. Regardless, separate normalizations would make these results difficult to interpret.
 - o While there was no global correlation between age effects in pre- and post-natal life, did specific genomic contexts (eg CGI shores, open chromatin regions, etc) and/or specific methylation states (hypo (0-0.2), partial/intermediary (0.2-0.8), and hyper (0.8-1) methylation levels, show more correlations? These comparisons do seem a bit susceptible to floor or ceiling effects, eg if pre-natal DNAm goes from 0.5 to 0, post-natal DNAm really can only increase (or stay steady) over aging.

- SATB2 nuclei sorting and pre- vs post-natal comparisons:

o What other cell type(s) are (or could be) present in the SATB2- nuclei that might confound the interpretation of these data? Our past work using Illumina 450k and cellular deconvolution on the developing and aging cortex (<https://pmc.ncbi.nlm.nih.gov/articles/PMC4783176/>) showed loss of pluripotency markers during a much narrower developmental window (14-20 pcw) and I would speculate that the cell type(s) underlying those "ESC" and "NPC" signatures are likely driving a lot of age-related signal in the current dataset. This seems further supported by Figure 4 as sorting only for SATB2 does not really capturing the full extent of cellular diversity in these prenatal samples. I think using the cell type-specific signatures from the smaller sorted samples to deconvolute the larger bulk samples would add some additional context to the current results, and teasing apart what cell type(s) are actually changing in pre-natal life would be an important contribution to the field.

o Were the two sorted datasets jointly processed and normalized, per the same comment above as combining the pre- and post-natal bulk datasets?

- Pre-natal cell type-specific DNAm patterns

o What extent of bulk prenatal DNAm changes can be attributed to the two extremes of (1) changes only in cell composition over age, with no association between DNAm levels and age within any cell type, versus (2) identical age-related changes occurring within each cell type? Supp Figure 14 suggests #2 might be more plausible but the low degree of age-related changes within each cell type might lean more towards #1 (although the sample sizes are admittedly smaller). Still, the lack of clarity around what other cell type are actually present in SATB2-nuclei makes these results difficult to interpret.

- Disease associations - what there any association between magnitude of DNAm change in development and ASD or SCZD association? The current analysis places equal weight on DMPs ranked 1st and 40000th but some degree of linearity would add additional weight to these findings.

Language/grammar/spelling:

- Bottom of page 4: "We fitted a linear model controlling for..." should probably clarify/explicitly state the linear/additive model: "We fitted a linear model of DNAm levels as a function of PCW, controlling for..."

Authors' response to the first round of review

Reviewer 1

1. Similar analysis has been done before more than 10 years ago in the Numata et al Am J Hum Genet 2012 paper. This paper used a cohort of samples with a similar age distribution and size. Therefore, it would be important to explain how the current paper improves on this previous studies (e.g. by analyzing additional developmental stages, using improved method etc).

We acknowledge that previous studies have investigated DNA methylation across human brain development; in the original submission we compared our results with data from a previous paper from our team (Spiers *et al.*)¹ that was published subsequent to the Numata *et al.*² study and currently represents the most systematic investigation of neurodevelopmental changes in DNA methylation. We agree that it is also important to highlight the Numata *et al.* paper mentioned by the reviewer and we apologise for not doing this in the original submission.

There are several improvements in terms of samples, experimental approach and analysis methods in our current study (compared to both the Numata *et al.* and Spiers *et al.* papers). First, the analyses in these previous publications were only performed on 'bulk' cortex tissue, while we also assess changes in purified populations SATB2+ and SATB2- nuclei. Second, the Numata paper used the Illumina 27K array to profile DNA methylation at 27,578 sites across the genome. In this study we use the Illumina EPIC array that quantifies DNA methylation at 866,150 sites across the genome (>31X as many sites as Numata et al). Third, the Numata paper explored DNA methylation in tissue from 108 donors (30 prenatal, 78 postnatal) whereas our study generated data from a total of 999 donors (112 prenatal, 887 postnatal). Furthermore, the range of ages profiled in the current study (6 post-conception weeks (pcw) to 108 years) is larger than those profiled in the Numata *et al.* paper (14 weeks post-conception to 84 years). Given the dramatic changes seen across our earliest fetal samples, the extended range of fetal cortex ages is a particular strength of the current study. Fourth, we have developed several analysis methods to analyse our data (including our approach for exploring non-line trajectories) and have incorporated novel annotation datasets that have been generated using novel methods since the Numata paper was published (e.g. single-cell ATAC-seq and RNA-seq data). We have also been able to leverage data from recent genetic studies of neurodevelopmental disorders (e.g. schizophrenia and autism) that have progressed considerably since 2012 via large collaborative association studies such as those performed by the Psychiatric Genomics Consortium.

The Reviewer's comment did prompt us to test for consistency between our data and that generated by Numata and colleagues (in the same way as we originally presented for the Spiers *et al.* data). As expected we found a very strong correlation in effect sizes for i) the DMPs reported in the Numata paper that are also included in our final dataset (n = 88 DMPs,

correlation = 0.92, $P = 1.6 \times 10^{-36}$) and ii) the DMPs identified in the current study for which results were available for in the Numata *et al.* paper ($n = 66$ DMPs, correlation = 0.94, $P = 2.8 \times 10^{-31}$, **Figure 1**). We have now included these results alongside the comparisons with the Spiers *et al.* dataset in the Supplementary Figures document.

Figure 1 - Effect sizes at dDMPs reported by Numata *et al.* (2012) are strongly correlated with DNA methylation changes at the same sites identified in the current study (corr = 0.94, $P = 2.8 \times 10^{-31}$).

2. Similarly, it would be important to clearly highlight the findings that are novel and were not shown by the Numata paper.

As described above, our study substantially extends and advances the work of Numata *et al.* in several important ways. First, we perform numerous novel analyses that were not possible at the time due to technological limitations. For example, our study uses genome-wide methylation data with far more extensive coverage compared to the 27K Illumina array used by Numata *et al.* Of these, only 24,282 sites (3.01%) overlap with those analyzed in the earlier study. As a result, just 855 (1.68%) of the 50,913 developmentally associated differentially methylated positions (dDMPs) identified in our bulk cortex data were even tested in the Numata *et al.* paper; the remaining 98.3% represent novel findings that could not have been assessed using

the 27K array. Additionally, our sample set spans a wider developmental age range than that of Numata *et al.*, which is particularly important for characterizing early fetal stages - periods during which we observe dramatic shifts in DNA methylation. Our study also examines DNA methylation changes in purified neuronal and non-neuronal populations, whereas Numata *et al.* only analyzed bulk cortex tissue, preventing any cell-type-specific insights. Moreover, the functional annotation resources we used - such as single-cell ATAC-seq datasets - were not available at the time of the earlier study. Finally, we demonstrate enrichments for genetic variants associated with autism and schizophrenia that were discovered after 2012, further emphasizing the novelty and contemporary relevance of our findings. Where appropriate we have clarified and emphasized these points in the revised manuscript.

3. If it is feasible, performing a meta-analysis of the Numata data with the current dataset would greatly improve the paper.

As noted above, only a small proportion (3.01%) of the CpG sites profiled in our current study were also assessed in the Numata *et al.* analysis, making a formal meta-analysis feasible for only a limited number of sites. Additionally, the Numata *et al.* paper reported results for just the top 99 dDMPs in their Supplementary Data file, and the link to the full dataset is no longer functional. To our knowledge, the raw methylation data are not publicly available, which further limits the possibility of conducting a meaningful meta-analysis. However, as described above, the reviewer's suggestion prompted us to assess concordance between our data and that of Numata *et al.* using the available results. Specifically, we examined effect size correlations for the 99 top-ranked dDMPs reported by Numata *et al.* that are also present in our dataset. Of these, 88 sites overlapped and showed a strong correlation (corr = 0.92, $P = 1.6 \times 10^{-36}$; see **Figure 1**). Similarly, for the 66 dDMPs identified in our study that were also reported in the Numata paper, we observed a correlation of 0.94 ($P = 2.8 \times 10^{-31}$). These comparisons are now included in the Supplementary Figures alongside those already presented for the Spiers *et al.* dataset. To further illustrate consistency across studies for the small number of overlapping sites, **Figure 2** presents DNA methylation trajectories at cg21006686 (annotated to *NOS1*), corresponding to Figure 3A in Numata *et al.* Our data recapitulate the developmental pattern reported by Numata *et al.*, while extending the age range substantially (6 weeks post-conception to 104 years vs. 14 weeks post-conception to 84 years) and incorporating a larger sample size.

Figure 2 - Consistent life-course patterns of DNA methylation at a site in *NOS1* in both **A)** the Numata et al. (2012) paper (figure corresponds to Fig 3A in that paper) and **B)** the current study. The current study includes a larger number of samples from a wider range of ages.

4. Is there a way to analyze CG and CH methylation using the data the authors produced? Given the importance and enrichment of CH methylation in neurons during postnatal development, this would be an important analysis.

This is an interesting suggestion by the Reviewer, and it prompted us to explore the extent of CH methylation in our bulk cortex data. Of note, the ability to explore CH methylation is constrained by the very small number of non-CG sites included on the Illumina EPIC array ($n = 1,277$; 0.158% of the sites included in our final dataset). In total, 8 CH sites show a significant change in DNAm across development when correcting for 1,277 sites, although only one of these sites is included in the list of 50,913 bulk cortex dDMPs passing our genome-wide significance threshold for all sites tested (ch.14.30061788F (not annotated to any gene), chromosome 14, $P = 4.39 \times 10^{-9}$) (**Figure 3**). Interestingly, all CH dDMPs are characterised by increasing DNAm across development and an analysis of 'total' CH methylation (derived by averaging across all 1,277 CH sites) highlights a small but significant increase across development (effect size = 0.0356% change in DNAm per week, $p = 0.00186$). We have added a brief description of these results to the manuscript and thank the Reviewer for this suggestion.

Figure 3 - The top-ranked non-CpG (CH) site is characterised by a significant increase in DNA methylation across human cortex development.

5. SATB2 sorting is used to enrich for neurons. However, SATB2 is enriched in upperlayer projection neurons in the cortex. Authors need to show what kind of population they are actually enriching.

We thank the reviewer for this important comment. As noted, SATB2 is a well-established marker of callosal projection neurons, which are predominantly located in the upper layers of the developing cortex. We fully acknowledge that SATB2 expression does not encompass all neuronal populations and have emphasized this limitation in the revised manuscript. To better reflect the specificity of our sorting strategy, we now use the term "SATB2-positive nuclei" rather than "neurons" where appropriate. Our original submission included snRNAseq data demonstrating that our SATB2-based FANS protocol enriches for a specific subpopulation of excitatory neurons, particularly upper-layer projection neurons, which are known to play crucial roles in cortical circuit formation and are highly relevant to the neurodevelopmental processes under investigation. In response to the reviewer's suggestion, we conducted an additional analysis using unpublished snRNA-seq data generated from bulk cortex tissue from a subset of our fetal cortex samples ($n = 38$ donors, 9 – 20pcw). Specifically, we quantified the proportion of different cell types among nuclei that do or do not express SATB2. As shown in **Figure 4**, this orthogonal approach confirms a clear enrichment of excitatory neurons among SATB2-expressing nuclei in the fetal cortex (94.9% of SATB2-expressing cells are excitatory neurons), while nuclei lacking detectable SATB2 expression are depleted for excitatory neurons (24.0% of

SATB2-non-expressing cells are excitatory neurons). While some other cell types show low levels of SATB2 expression, and some excitatory neurons lack detectable SATB2, this is consistent with the known sparsity and dropout issues in single-nucleus RNA-seq data. We have added some discussion to the manuscript to highlight the cell-type-specificity of SATB2 expression and clarified the language used in the manuscript.

Figure 4 - Cell-type proportions in sub-populations of cells found to express or not express SATB2 in unpublished snRNA-seq data derived from a subset of the fetal cortex samples used in this study.

6. It is not clear whether all p values were corrected for multiple comparisons. This needs to be clearly indicated everywhere (corrected or uncorrected p value).

Yes, all P values are corrected for multiple testing. We believe this was quite clearly stated in the original manuscript - e.g. when introducing our dDMP results we note “...at an empirically-derived experiment-wide significance threshold ($p < 9 \times 10^{-8}$)...” and include a citation to our previous work that justifies this threshold as a robust threshold for epigenomewide association studies. Other statistical tests use appropriate corrections, and we have double-checked the manuscript to ensure these are clearly stated in either the methods or results as appropriate.

Reviewer 2

1. Prenatal age associations: these analyses are well described and easy to follow. However, I would probably integrate the non-linear and WGCNA modeling into the first results subsection describing age-related changes during pre-natal life since they generally support the extensive DNAm changes occurring during this critical period. Then the authors should perform analogous enrichment analyses as the linear changes (to more active genomic states) for CpGs with non-linear changes and with various WGCNA clusters to help better interpret/contextualize them.

In response to the Reviewer's comment, we have revised the structure of the manuscript to more clearly integrate the findings from both linear and non-linear DNA methylation changes. We agree that these modifications significantly improve the clarity and flow of the Results section. To strengthen the comparison between linear and non-linear sites, we also performed additional analyses of the non-linear data. Notably, we found that non-linear sites exhibit a similar pattern of enrichment within cell type-specific ATAC-seq peaks as observed for linear sites. However, individual non-linear WGCNA modules display enrichment for distinct cell types, suggesting potential differences in their functional relevance (**Figure 5**). These additional analyses have been briefly described in the revised manuscript.

Figure 5 - Heatmap showing relative enrichment within cell-type-specific ATAC-seq peaks of linear and non-linear dDMPs (left) and non-linear sites within individual WCGNA modules (right).

2. It doesn't look like the pre- and post-natal DNAm samples were generated in the same batches, which might limit direct comparisons of variance (eg Figure 2A). Can the authors clarify if these samples were at least processed+normalized together with dasen for these analyses?

We thank the reviewer for this observation. The postnatal cortex data were generated as part of a separate study and therefore processed independently from the prenatal samples. While we acknowledge that this could potentially introduce subtle technical variation in our secondary analyses comparing fetal and postnatal cortex, we note that all data were processed and normalized using the same pipeline. We would argue that because our models were applied *within* each dataset (with the subsequent comparison of effect sizes for age), it minimises the risk of technical confounding in the identification of differences that might occur if two different batches of data were pooled.

Importantly, the dramatic increase in variance at dDMPs among fetal cortex samples compared to postnatal cortex samples are specific to dDMPs and not observed across the genome. Furthermore, the effect sizes observed in the fetal cortex far exceed anything that would result

from batch effects - many of these sites are characterised by a shift in DNA methylation status from near 0% to 100% (or vice versa). To further address this concern, we now include additional analyses demonstrating that among an equal number of lowranked probes (i.e. the bottom 10,000 ranked dDMPs), variance is comparable between fetal and postnatal samples (**Figure 6**); in fact, there is slightly more in variance among postnatal samples at these sites as would be expected given the much larger number of samples and the much wider timescales profiled. We also explored DNA methylation levels across probes in the promoter regions of canonically expressed housekeeping genes; we found these to be stable with no differences in DNA methylation levels (or variance) between prenatal and postnatal cortex (see **Figure 7** for examples of sites in the promoter of GAPDH and ACTB). These results argue against a systemic batch-driven difference in DNA methylation or variance and support our interpretation that increased variance at dDMPs reflects meaningful biological heterogeneity during fetal development. Nonetheless, we now explicitly acknowledge in the manuscript that independent processing of prenatal and postnatal samples is a limitation when making direct comparisons between these datasets.

Figure 6 – Distribution of variance statistics in fetal and adult samples for the 10,000 highest-ranked sites for developmental effects (i.e. top 10,000 dDMPs) (left) and the 10,000 lowest-ranked DNA methylation sites for developmental effects (right).

Figure 7 – Sites annotated to canonically expressed housekeeping genes *GAPDH* (top) and *ACTB* (bottom) show consistent and stable levels of DNA methylation across the life-course with no difference between prenatal and postnatal cortex samples.

3. While there was no global correlation between age effects in pre- and post-natal life, did specific genomic contexts (eg CGI shores, open chromatin regions, etc) and/or specific methylation states (hypo (0-0.2), partial/intermediary (0.2-0.8), and hyper (0.8-1) methylation levels, show more correlations? These comparisons do seem a bit susceptible to floor or ceiling effects, eg if pre-natal DNAm goes from 0.5 to 0, post-natal DNAm really can only increase (or stay steady) over aging.

We thank the reviewer for this insightful comment, and agree that there is the *potential* for ‘ceiling effects’ in the comparison of changes in pre- and post-natal samples; we have noted this as a caveat in the revised manuscript. The Reviewer’s comments prompted us to perform additional analyses to compare the correlation of age-associated DNA methylation changes in prenatal and postnatal cortex at sites located in different genomic contexts. In general, we observe limited correlation in age effects between the prenatal and postnatal cortex, consistent with our earlier analysis of all sites (see **Figure 8**). Specific regions, however, were characterised by more consistent patterns. Notably, 12% of dDMPs located within CpG islands also show age-associated methylation changes in the same direction in postnatal cortex (**Figure 8**). Interestingly, while dDMPs overall are significantly depleted in CpG islands, age-

associated changes in the postnatal cortex are significantly enriched in these regions (**Figure 9**). Together, these findings suggest that CpG islands may be preferentially involved in general aging-related methylation changes, whereas dDMPs in other genomic contexts (e.g., shores and shelves) are more likely to reflect development-specific epigenetic remodeling. These results concur with a) data from previous studies showing that dynamic methylation changes during early brain development occur largely outside of CpG islands², and b) data showing that age-associated methylation changes are enriched in CpG islands, suggesting a consistent aging signature in these regions³. We have included discussion of these results in the revised manuscript.

Figure 8 - Comparison of age effect sizes between prenatal and postnatal cortex for the 41,518 dDMPs also measured in the postnatal samples, split by their overlap with CpG feature.

Figure 9 - Comparison of CpG and genic feature enrichment in prenatal development-associated DMPs and postnatal age-associated DMPs. Left) Enrichment of prenatal dDMPs in CpG and genic features. CpG islands (CGI) show a significant depletion (1.65% of dDMPs are annotated to CGI, $p < 1 \times 10^{-320}$). Right) Of the dDMPs that are also significant postnatally ($n = 1,003$) there is a significant enriched within CGI (8.86% of postnatal DMPs, $p = 3.97 \times 10^{-99}$).

4. What other cell type(s) are (or could be) present in the SATB2- nuclei that might confound the interpretation of these data? Our past work using Illumina 450k and cellular deconvolution on the developing and aging cortex (<https://pmc.ncbi.nlm.nih.gov/articles/PMC4783176/>) showed loss of pluripotency markers during a much narrower developmental window (14-20 pcw) and I would speculate that the cell type(s) underlying those "ESC" and "NPC" signatures are likely driving a lot of age-related signal in the current dataset. This seems further supported by Figure 4 as sorting only for SATB2 does not really capturing the full extent of cellular diversity in these prenatal samples. I think using the cell type-specific signatures from the smaller sorted samples to deconvolute the larger bulk samples would add some additional context to the current results, and teasing apart what cell type(s) are actually changing in pre-natal life would be an important contribution to the field.

The Reviewer raises an important and valid point. As the SATB2- population is not positively sorted, it is indeed more heterogeneous than the SATB2+ population and likely comprises a mixture of diverse cell types. As noted in our response to Reviewer 1 (point 5), our laboratory has generated unpublished single-nucleus RNA-seq (snRNA-seq) data from an overlapping set of samples. This dataset enables us to interrogate the cellular composition of both SATB2-expressing and non-expressing nuclei (see **Figure 4**). We now explicitly acknowledge in the revised manuscript that isolating nuclei solely on the basis of SATB2 expression does not

capture the full spectrum of cellular diversity in the developing cortex, and we highlight this as a limitation of the approach.

In response to the Reviewer's suggestion, we additionally performed cell-type deconvolution on our bulk cortical DNA methylation data spanning early to mid-fetal development. To do this, we leveraged reference profiles from SATB2+, NeuN+, and NeuN- sorted nuclei obtained from late fetal and early postnatal human cortex, as well as embryonic stem cell (ESC) and neural progenitor cell (NPC) signatures previously described in Kim *et al.* ⁴, and used by the Reviewer in their own prior work ⁵. Deconvolution was performed using the CETYGO package, as previously published by our team ⁶. As shown in **Figure 10**, this analysis reveals a striking developmental increase in the proportion of SATB2+ cells over time. Notably, this trajectory closely mirrors the increase in SATB2+ nuclei observed in our snRNA-seq dataset from a subset of the same samples (**Figure 11**), providing orthogonal validation. Concurrently, we observe a decrease in the estimated proportion of ESC-like cells across this period, consistent with the Reviewer's hypothesis regarding developmental progression. In contrast, estimated proportions of NeuN+ nuclei remain minimal throughout this window, corroborating both our FANS data and transcriptomic analyses that demonstrate low NeuN expression during this developmental stage. These findings reinforce our conclusion that NeuN is not a reliable marker for identifying neurons in the early developing human cortex. We thank the Reviewer for this insightful suggestion and have included a summary of these results in the revised version.

Figure 10 – Predicted cell-type proportions across early- and mid-fetal cortex samples derived using reference DNA methylation data. ES = embryonic stem cells (ES) ⁴. ES_NPC = ES-derived neural precursor cells ⁴. SATB2pos = SATB2+ late-fetal (20 – 28pcw) and early postnatal (0 – 8y) samples (this study). NeuNpos = postnatal NeuN+ samples (this study). NeuNneg = postnatal NeuN- samples (this study).

Figure 11 – Change in excitatory and inhibitory neuron proportions with fetal age in unpublished snRNA-seq data derived from a subset of the fetal cortex samples used in this study. The changes in excitatory neuron proportion parallels the changes in SATB2+ proportion derived from the DNA methylation data in **Figure 10**.

5. Were the two sorted datasets jointly processed and normalized, per the same comment above as combining the pre- and post-natal bulk datasets?

As noted above in our response to comment 2, the prenatal and postnatal cortex samples were processed separately, as the postnatal data were generated previously by our group as part of ongoing analyses related to another project. We used this existing dataset to examine the temporal specificity of the developmental changes observed in the prenatal cortex. Combining the datasets would not be appropriate given the independent nature of each dataset. Therefore, while the datasets were not jointly normalized, both were processed in the same way, and within each dataset, neuronal and non-neuronal nuclei were processed together. This ensures that comparisons between cell types are internally consistent and not confounded by batch effects. As with our bulk cortex analyses, the magnitude and specificity of the developmental changes observed - particularly the large shifts in methylation at dDMPs - greatly exceed what would be expected from technical variation. We further show that sites annotated to promoter regions of housekeeping genes exhibit consistent methylation levels and variance in each cell population across datasets (**Figure 12**), supporting the conclusion that our findings reflect genuine biological changes rather than artefactual differences arising from separate processing.

Figure 12 - Sites annotated to canonically expressed housekeeping genes *GAPDH* (top) and *ACTB* (bottom) remain stably methylated across FANS cortex age groups in both *SATB2+* and *SATB2-* populations.

6. What extent of bulk prenatal DNAm changes can be attributed to the two extremes of (1) changes only in cell composition over age, with no association between DNAm levels and age within any cell type, versus (2) identical age-related changes occurring within each cell type? Supp Figure 14 suggests #2 might be more plausible but the low degree of age-related changes within each cell type might lean more towards #1 (although the sample sizes are admittedly smaller).

We thank the Reviewer for this comment. We agree that many of the developmental DNA methylation changes observed appear to be broadly consistent across the two cell populations profiled in the second part of our study. The relatively small number of statistically significant differentially methylated positions (dDMPs) identified in the sorted populations is likely attributable to reduced statistical power, owing to the substantially smaller sample sizes compared to our bulk cortex analyses. This interpretation is supported by the strong correlation in effect sizes at bulk cortex dDMPs when comparing the bulk tissue to each of the sorted nuclei populations presented in our original submission. Despite the limited power, we nonetheless observe some cell-type-specific patterns, with a subset of loci exhibiting divergent

developmental trajectories between the sorted populations indicating that cell-type-specific methylation dynamics do exist and can be detected. The Reviewer's suggestion prompted us to further evaluate whether age-related changes in celltype composition could confound our DNA methylation findings. To assess this, we implemented two complementary approaches: a) We estimated the proportion of SATB2+ nuclei in each bulk cortex sample using our cell-type deconvolution framework applied to DNA methylation data, as described in our response to Comment 4; b) We also derived independent estimates of the proportion of excitatory (SATB2+) neurons using the matched, unpublished single-nucleus RNA-seq (snRNA-seq) data from an overlapping set of samples. These estimates were extrapolated from the cell-type proportions shown in **Figure 11**. We then incorporated each of these SATB2+ cell proportion estimates as covariates in revised linear models assessing DNA methylation trajectories. As shown in **Figure 13**, which

Figure 13 – Effect sizes for the top 100 dDMPs identified in the bulk fetal cortex remain largely unchanged before and after correcting for SATB2+ proportion derived from DNA methylation data (left) and excitatory neuron proportions derived from snRNA-seq data (right).

compares effect sizes for the top 100-ranked dDMPs from the original model with those derived from models adjusted for SATB2+ cell proportions (left: DNA methylation-derived; right: snRNA-seq-derived), the inclusion of these covariates had minimal impact on the results. This suggests that developmental changes in cell-type composition have a negligible effect on our observed dDMPs. We do, however, now include some discussion about the potential confounding resulting from variable cell-type proportions.

7. Disease associations - what there any association between magnitude of DNAm change in development and ASD or SCZD association? The current analysis places equal weight on DMPs ranked 1st and 40000th but some degree of linearity would add additional weight to these findings.

We thank the Reviewer for this excellent suggestion. In our original analyses assessing the enrichment of SFARI autism genes and SCHEMA schizophrenia loci among genes annotated to dDMPs, we weighted all significant dDMPs equally. In response to the Reviewer's comment, we performed a tiered enrichment analysis across sequential association bins, starting with the top-ranked dDMPs and progressively including less strongly associated sites. Notably, we observed that the top-ranked dDMPs showed the strongest enrichment for disease-associated genes, consistent with the Reviewer's hypothesis, in both bulk-cortex and SATB2+ neurons (see **Figure 14** for bulk cortex enrichments across ranked dDMP bins). We have added this new analysis as a Supplementary Figure in the revised manuscript.

Figure 14 - Enrichment of dDMPs in SFARI and SCHEMA genes is strongest among highest-ranked dDMPs.

8. Bottom of page 4: "We fitted a linear model controlling for..." should probably clarify/explicitly state the linear/additive model: "We fitted a linear model of DNAm levels as a function of PCW, controlling for..."

We have amended the wording as per the Reviewer's suggestion.

References

1. Spiers, H., Hannon, E., Schalkwyk, L.C., Smith, R., Wong, C.C.Y., O'Donovan, M.C., Bray, N.J., and Mill, J. (2015). Methylomic trajectories across human fetal brain development. *Genome Res* 25, 338–352. <https://doi.org/10.1101/GR.180273.114>.
 2. Numata, S., Ye, T., Hyde, T.M., Guitart-Navarro, X., Tao, R., Winger, M., Colantuoni, C., Weinberger, D.R., Kleinman, J.E., and Lipska, B.K. (2012). DNA Methylation Signatures in Development and Aging of the Human Prefrontal Cortex. *Am J Hum Genet* 90, 260. <https://doi.org/10.1016/J.AJHG.2011.12.020>.
 3. Horvath, S. (2013). DNA methylation age of human tissues and cell types. *Genome Biol* 14, 1–20. <https://doi.org/https://doi.org/10.1186/gb-2013-14-10-r115>.
 4. Kim, M., Park, Y.K., Kang, T.W., Lee, S.H., Rhee, Y.H., Park, J.L., Kim, H.J., Lee, D., Lee, D., Kim, S.Y., et al. (2014). Dynamic changes in DNA methylation and hydroxymethylation when hES cells undergo differentiation toward a neuronal lineage. *Hum Mol Genet* 23, 657–667. <https://doi.org/10.1093/HMG/DDT453>.
 5. Jaffe, A.E., Gao, Y., Deep-Soboslay, A., Tao, R., Hyde, T.M., Weinberger, D.R., and Kleinman, J.E. (2015). Mapping DNA methylation across development, genotype, and schizophrenia in the human frontal cortex. *Nat Neurosci* 19, 40. <https://doi.org/10.1038/NN.4181>.
 6. Vellame, D.S., Shireby, G., MacCalman, A., Dempster, E.L., Burrage, J., GorrieStone, T., Schalkwyk, L.S., Mill, J., and Hannon, E. (2023). Uncertainty quantification of reference-based cellular deconvolution algorithms. *Epigenetics* 18. <https://doi.org/10.1080/15592294.2022.2137659>.
-

Referees' reports, second round of review

Reviewer #2:

The authors comprehensively answered my previous questions. I think the only minor issue relates to the SATB2 sorting and subsequent DNA methylation analyses. In the Review Response Figure 4, while the SATB2+ nuclei did show an expected enrichment of excitatory neurons, the SATB2- nuclei appeared predominantly neuronal, eg Excitatory + Inhibitory fractions > 65%, by this approach. During the actual cell sorting, what fraction of nuclei were positively selected (eg SATB2+)? I think it would have to be a small percentage, like <10%, for the results in the paper to make sense, eg having large DNAm changes between SATB2+ and SATB2- populations. I also think based on this Review Response Figure 4, can the authors really call this population "non-neuronal-enriched" when its predominantly neuronal? I think the text in this results subsection should be clarified.

Authors' response to the second round of review

Reviewer 2

The authors comprehensively answered my previous questions. I think the only minor issue relates to the SATB2 sorting and subsequent DNA methylation analyses. In the Review Response Figure 4, while the SATB2+ nuclei did show an expected enrichment of excitatory neurons, the SATB2- nuclei appeared predominantly neuronal, eg Excitatory + Inhibitory fractions > 65%, by this approach. During the actual cell sorting, what fraction of nuclei were positively selected (eg SATB2+)? I think it would have to be a small percentage, like <10%, for the results in the paper to make sense, eg having large DNAm changes between SATB2+ and SATB2- populations. I also think based on this Review Response Figure 4, can the authors really call this population "non-neuronal-enriched" when its predominantly neuronal? I think the text in this results subsection should be clarified.

Thank you for this thoughtful comment. We agree that the terminology used in the original manuscript to describe the SATB2- population was imprecise. In response, we have revised the text to describe this population as "*neuron-depleted*" rather than "*non-neuronalenriched*", which more accurately reflects the composition and intent of the sorting strategy. Of note, SATB2+ nuclei comprised approximately 30% of total DAPI+ nuclei isolated from fetal cortex.

We also acknowledge the Reviewer's query regarding the cellular composition shown in Review Response Figure 4. We apologise for any confusion – as stated in the response text, this figure does not directly represent the sorted SATB2- nuclei population used for DNA methylation analysis. Instead, it was generated from snRNA-seq data of unsorted bulk fetal cortex, and was intended to illustrate the cell-type specificity of SATB2 expression in this tissue. As the figure shows, SATB2-expressing nuclei are highly enriched for excitatory neurons, whereas nuclei in which SATB2 is not detected are relatively depleted of these cells. However, due to the sparsity and dropout inherent to single-nucleus RNA-seq some excitatory neurons will inevitably appear to not-express SATB2 in these analyses.

Finally, as the Reviewer notes, we do expect some neurons - particularly inhibitory neurons - in the SATB2- population, as SATB2 is not robustly expressed in these cells. We have revised the relevant section of the results to more clearly describe the heterogeneity of the SATB2- population and to explicitly state in the discussion that while this group is depleted of excitatory neurons, it is not devoid of neuronal cells.